# Interferon-α promotes HLA-B-restricted presentation of conventional and alternative antigens in human pancreatic β-cells

Alexia Carré[1], Fatoumata Samassa[1,12], Zhicheng Zhou[1,12], Javier Perez-Hernandez[1,2], Christiana Lekka[3], Anthony Manganaro[4], Masaya Oshima[1], Hanqing Liao[5], Robert Parker[5], Annalisa Nicastri[5], Barbara Brandao[1], Maikel L. Colli[6], Decio L. Eizirik[6], Jahnavi Aluri[7], Deep Patel[7], Marcus Göransson[8], Orlando Burgos Morales[1], Amanda Anderson[9], Laurie Landry[9], Farah Kobaisi[1], Raphael Scharfmann[1], Lorella Marselli[10], Piero Marchetti[10], Sylvaine You[1,7], Maki Nakayama[9], Sine R. Hadrup[8], Sally C. Kent[4], Sarah J. Richardson[3], Nicola Ternette[5] & Roberto Mallone[1,7,11] ✉

Interferon (IFN)-α is the earliest cytokine signature observed in individuals at risk for type 1 diabetes (T1D), but the effect of IFN-α on the antigen repertoire of HLA Class I (HLA-I) in pancreatic β-cells is unknown. Here we characterize the HLA-I antigen presentation in resting and IFN-α-exposed β-cells and find that IFN-α increases HLA-I expression and expands peptide repertoire to those derived from alternative mRNA splicing, protein *cis*-splicing and post-translational modifications. While the resting β-cell immunopeptidome is dominated by HLA-A-restricted peptides, IFN-α largely favors HLA-B and only marginally upregulates HLA-A, translating into increased HLA-B-restricted peptide presentation and activation of HLA-B-restricted CD8+ T cells. Lastly, islets of patients with T1D show preferential HLA-B hyper-expression when compared with non-diabetic donors, and islet-infiltrating CD8+ T cells reactive to HLA-B-restricted granule peptides are found in T1D donors. Thus, the inflammatory milieu of insulitis may skew the autoimmune response toward alternative epitopes presented by HLA-B, hence recruiting T cells with a distinct repertoire that may be relevant to T1D pathogenesis.

Type I interferons (IFNs), notably IFN-α and IFN-β, play a crucial role in anti-viral immune responses by driving infected cells to express IFN-stimulated genes, thus limiting viral replication and spreading[1]. Type I IFNs are also central in modulating innate immune responses and activating adaptive immunity[1]. They boost B-cell differentiation and antibody (Ab) production, mature antigen-presenting cells, and upregulate their expression of human leukocyte antigen (HLA) Class I (HLA-I) along with costimulatory molecules[1]. Upon detection of microbial components, macrophages and dendritic cells (DCs; mainly plasmacytoid DCs) secrete IFN-α, leading to additional cytokine and chemokine release. In concert with IFN-γ, mainly released by T cells, IFN-α also increases HLA-I expression directly on infected cells[2], thus augmenting their antigenic visibility, and enhances the cytolytic activity and survival of natural killer (NK) and CD8+ T cells[3].

Besides these potent anti-viral effects, IFNs can also promote autoimmunity[4]. In type 1 diabetes (T1D), transcriptomics investigations on the blood of children genetically at risk revealed a strong type I IFN signature preceding seroconversion[5,6]. At the tissue level, IFN-

stimulated genes are overexpressed only in infiltrated islets of recent-onset T1D patients[7], in line with an enhanced pancreatic IFN-α expression[8]. These IFN signatures may reflect exposure to viral infections, with Coxsackieviruses B proposed as possible environmental triggers of islet autoimmunity[9,10] or to other "danger signals".

IFN-α[8] and IFN-stimulated gene expression[11] colocalize with HLA-I hyper-expression, which is a histopathological hallmark of T1D in insulin (INS)-containing islets[12]. More recently, a causal relationship between IFN-α, endoplasmic reticulum (ER) stress, and HLA-I upregulation has been demonstrated in β-cells[2,13]. The increased surface HLA-I expression on β-cells may be a driver of T1D pathogenesis, as self-reactive CD8+ T cells must recognize peptide-HLA-I (pHLA-I) complexes on β-cells to trigger lysis. In addition, both IFN-α and ER stress may impact the repertoire of HLA-I-presented peptides (so-called immunopeptidome), as is the case for IFN-γ[14,15]. Indeed, IFN-α promotes the generation of mRNA splice variants[13,16], which can result in the presentation of alternative antigens spanning novel splicing sites. This loss of immune ignorance may also rely on other IFN-induced pathways of alternative antigen formation[14,17], e.g., post-translation modifications (PTMs), alternative transcription start sites generating an INS defective ribosomal product (DRiP)[18] and peptide cis-splicing, i.e., the fusion of non-contiguous protein fragments. Collectively, the non-canonical peptide sequences, i.e., not templated in the genome, resulting from these processes will be referred to as alternative antigens (also termed "neo-antigens" in the literature). These alternative antigens may be recognized by T cells as "non-self" and escape immune tolerance. Despite their potential relevance, evidence for their natural processing and presentation is missing in most cases[19].

Since IFN-α may modulate the antigenic cargo offered to T cells, elucidating how IFN-α shapes the immunopeptidome of β-cells is crucial to further understanding the antigenic drivers of T1D. To address this, we here apply immunopeptidomics strategies to human β-cells treated or not with IFN-α. We report that IFN-α skews antigen presentation in favor of HLA-B-restricted peptides, including both conventional and alternative antigens, which are recognized by islet-infiltrating CD8+ T cells of T1D patients. This work reveals an unexplored antigenic repertoire that may drive β-cell autoimmunity.

## Results
### ECN90 β-cells exposed to IFN-α increase surface HLA-I expression and peptide presentation
To understand immunopeptidome changes in β-cells exposed to IFN-α, we cultured the ECN90 β-cell line[20] with or without IFN-α for 18 h. pHLA-I complexes were immunoprecipitated from cell lysates, and peptides were identified by liquid chromatography-tandem mass spectrometry (LC-MS/MS). IFN-α-treated ECN90 β-cells yielded more peptides than untreated cells (Fig. 1a, b), with respectively 91.1% and 74.9% sequences in the expected 8–14 amino-acid (aa) range for HLA-I ligands (Fig. 1a, e). This increased peptide breadth reflected an IFN-α-induced upregulation of surface HLA-I (Fig. 1c, d). Unsupervised Gibbs clustering uncovered the peptide motifs of HLA-I alleles expressed by ECN90 β-cells (Supplementary Fig. 1): HLA-A*03:01 (HLA-A3 from hereon), group 1; HLA-A*02:01 (HLA-A2), -C*03:04 (-C3), -C*07:01 (-C7), group 2; HLA-B*40:01 (-B40), -B*49:01 (-B49), group 3.

Peptides were identified and filtered by a multi-step bioinformatics pipeline (Fig. 1e). The Swiss-Prot reference database used for the identification of genome-templated sequences was complemented with translated mRNA isoforms enriched in human islets exposed or not to IFN-α[13] and generating alternative aa sequences; and with INS open-reading frames covering published alternative epitopes[18]. Following selection for 8–14 aa length, and identification of mRNA variants, all sequences underwent filtering based on enriched expression of source proteins in β-cells. Peptides carrying PTMs were analyzed separately. All those spectra whose peptide interpretation did not match genome-templated, RNA-spliced, or PTM sequences were searched for putative proteasomal cis-spliced peptides using the MARS algorithm[21]. In total, this workflow led to the identification of 4 mRNA splice candidates (Supplementary Data 1), 784 conventional peptides (Supplementary Data 2), 247 sequences carrying PTMs (Supplementary Data 3), 11 cis-spliced sequences (Supplementary Data 4). HLA-I restrictions were then predicted using NetMHCpan 4.1a[22]. For all peptide categories, more HLA-I binders were identified in cells exposed to IFN-α (Fig. 1e), in line with the induced surface HLA-I upregulation.

Collectively, these results show that IFN-α increases surface HLA-I expression and peptide presentation.

### The immunopeptidome of ECN90 β-cells is dominated by insulin granule proteins and its breadth is increased by IFN-α
Building on this observation of increased peptide breadth induced by IFN-α, we set forth to identify the features of this expanded immunopeptidome. We started by investigating the source proteins of the β-cell-enriched peptides identified. IFN-α treatment increased not only the number of peptides presented but also the breadth of source proteins, with 35.6% (58/163) of them appearing only in IFN-α-treated cells as compared to 6.1% (10/163) specific to untreated cells (Supplementary Fig. 2a). Accordingly, 19 source proteins in IFN-α-treated cells yielded ≥ 2-fold more peptides ($\log_2$ fold change FC > 1) than in the control condition (Fig. 2a and Supplementary Fig. 2a). Conversely, only 8 source proteins displayed a similar $\log_2$FC < 1 in untreated vs. IFN-α-treated cells. In both conditions, the 5 source proteins yielding more peptides either localize in secretory granules, i.e., chromogranin A (CHGA, n = 132), INS (n = 64), secretogranin 5 (SCG5, n = 34); or are associated with granule transport, i.e., kinesin-like protein (KIF1A, n = 34) and microtubule-associated protein 1B (MAP1B, n = 26). Conversely, some known β-cell antigens were underrepresented (Supplementary Fig. 2a), i.e., zinc transporter 8 (SLC30A8, n = 5), glutamate decarboxylase (GAD2, n = 3) and islet amyloid polypeptide (IAPP, n = 1), or completely absent, i.e., islet-specific glucose-6-phosphatase catalytic subunit-related protein (IGRP) and INS DRiP[18]. Novel granule proteins were also identified, i.e., chromogranin B (CHGB, n = 6), secretogranin 2 and 3 (SCG2, n = 11; SCG3, n = 10).

To validate the results obtained on ECN90 β-cells, HLA-I-bound peptides were identified from two untreated HLA-A2/A3+ primary human islet preparations from non-diabetic donors. Despite the limited (~100-fold lower) amount of starting material and the dilution of β-cell-derived peptides with those from other endocrine cells, 342 β-cell-enriched conventional peptides were retrieved from these islet preparations (Supplementary Data 5), i.e., only 2-fold less than in ECN90 β-cells (n = 784). Across the 2 islet samples, the source proteins (n = 72) of the peptides retrieved overlapped by 44% with those from ECN90 β-cells (n = 163) (Supplementary Fig. 2b). Interestingly, even though glucagon (GCG)-secreting α-cells account for only half the number of β-cells in human islets[23], the most represented endocrine-enriched protein was GCG (n = 26 and 30 peptides from islet sample 1 and 2, respectively; n = 43 unique peptides; Supplementary Fig. 2c). The other dominant source proteins were otherwise the same as in ECN90 β-cells, notably CHGA (n = 7 + 10, 12 unique peptides) and INS (n = 29 + 24, 33 unique peptides; Fig. 2b and Supplementary Data 5). In total, 40/117 (34%) HLA-A2/-A3-restricted peptides identified in primary islets were also found in ECN90 β-cells (Supplementary Data 6), which was only slightly lower than the overlap between the two islet preparations (56/117 shared peptides, 48%).

Collectively, these results show that IFN-α increases not only the number of peptides displayed by HLA-I but also their breadth in terms of source proteins. Granule-contained and granule-associated proteins are abundantly represented, with some novel ones (CHGB, SCG2, SCG3) identified.

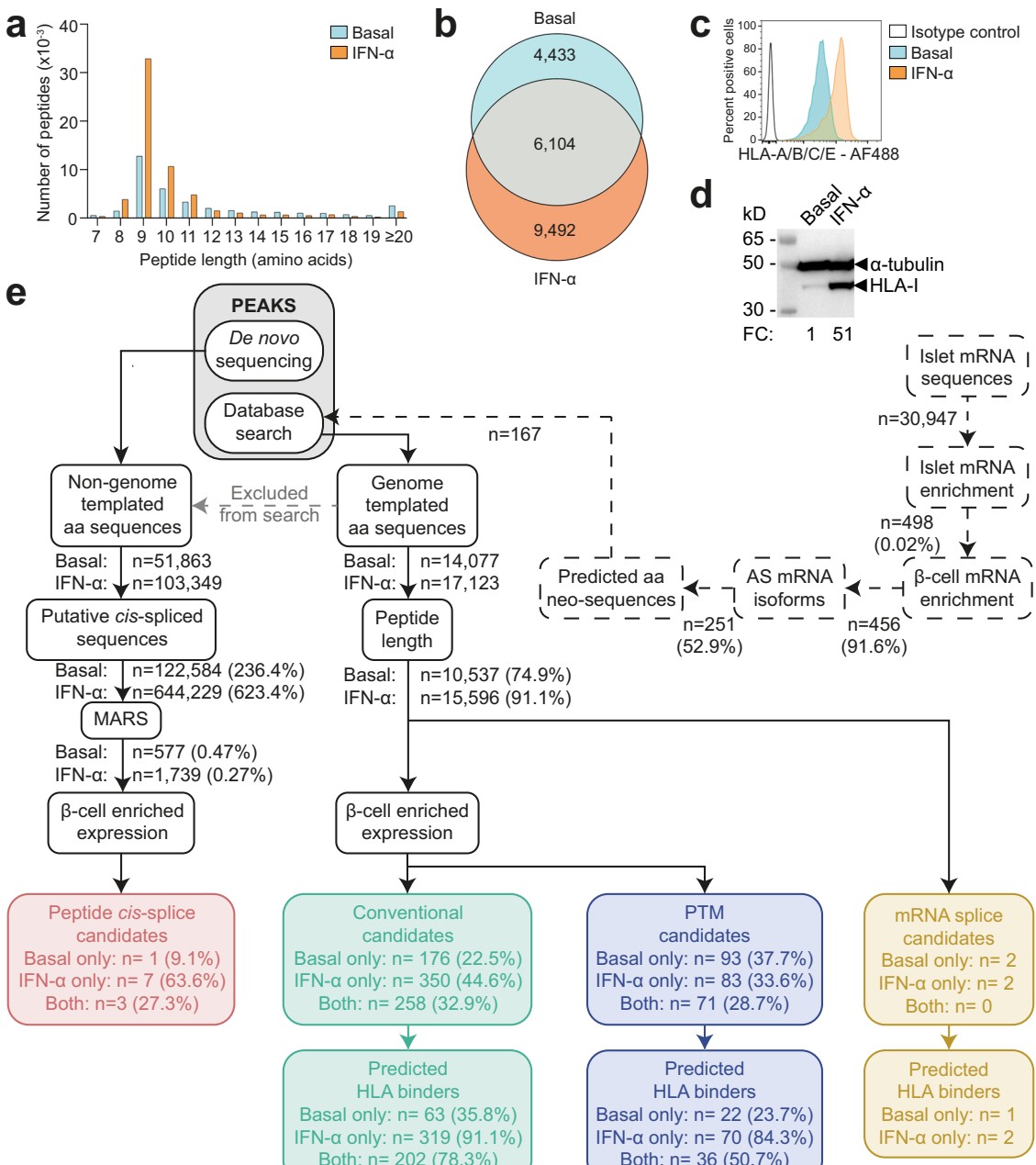

**Fig. 1 | Immunopeptidome profiling of HLA-I-bound peptides presented by ECN90 β-cells. a** Length distribution of HLA-I-eluted peptides from ECN90 β-cells under basal and IFN-α-stimulated conditions. Four biological replicates (3.5-4.5 × 10⁸ cells/each) were acquired for each condition, and unique peptides across replicates were counted. Bars represent cumulative peptide counts from all 4 replicates. **b** Number of 8-14mer peptides identified in at least one replicate of the basal, IFN-α-treated, or both conditions. Peptides found under basal conditions in one replicate and in both conditions in another were classified as "both". **c** HLA-I expression detected by flow cytometry on ECN90 β-cells in basal and IFN-α-stimulated conditions using W6/32 Ab. **d** HLA-I heavy chain expression detected with HC10 Ab by Western blot in whole cell lysates of ECN90 cells β-exposed or not to IFN-α, with α-tubulin bands as loading controls and normalized HLA-I fold change (FC) values indicated. For panels (**c**, **d**), representative experiments out of 4

performed are shown. **e** Bioinformatics analysis pipeline. Predicted alternative aa sequences from RNAseq datasets of human islets (dashed lines) were appended to the database used for immunopeptidome search. Database-matched sequences identified by PEAKS (gray box) were sequentially filtered based on their length, on whether they matched mRNA variants (yellow boxes; peptides listed in Supplementary Data 1), and on the enriched expression of their source proteins in β-cells. Conventional candidates (green boxes; Supplementary Data 2) and sequences carrying PTMs (violet boxes; Supplementary Data 3) were separated. HLA-I-binding predictions were performed using NetMHCpan4.1a (peptide motifs detailed in Supplementary Fig. 1). In parallel, non-genome-templated peptides were interpreted as potential *cis*-spliced candidates and fed into the MARS algorithm, followed by filtering according to the enrichment of their source proteins in β-cells (red box, Supplementary Data 4).

## The immunopeptidome of ECN90 β-cells is shaped by the chymotrypsin-like activity of both the constitutive and immunoproteasome

We pursued our studies by investigating the pathways that modify the immunopeptidome upon IFN-α treatment. Interestingly, IFN-α increased

the number and frequency of peptides with a chymotryptic motif (Fig. 2c, d). Since the immunoproteasome harbors 2 chymotrypsin-like catalytic subunits while the constitutive proteasome has only one[24], this pattern could reflect an increased immunoproteasome activity. We, therefore, explored the proteasome dependency of the identified

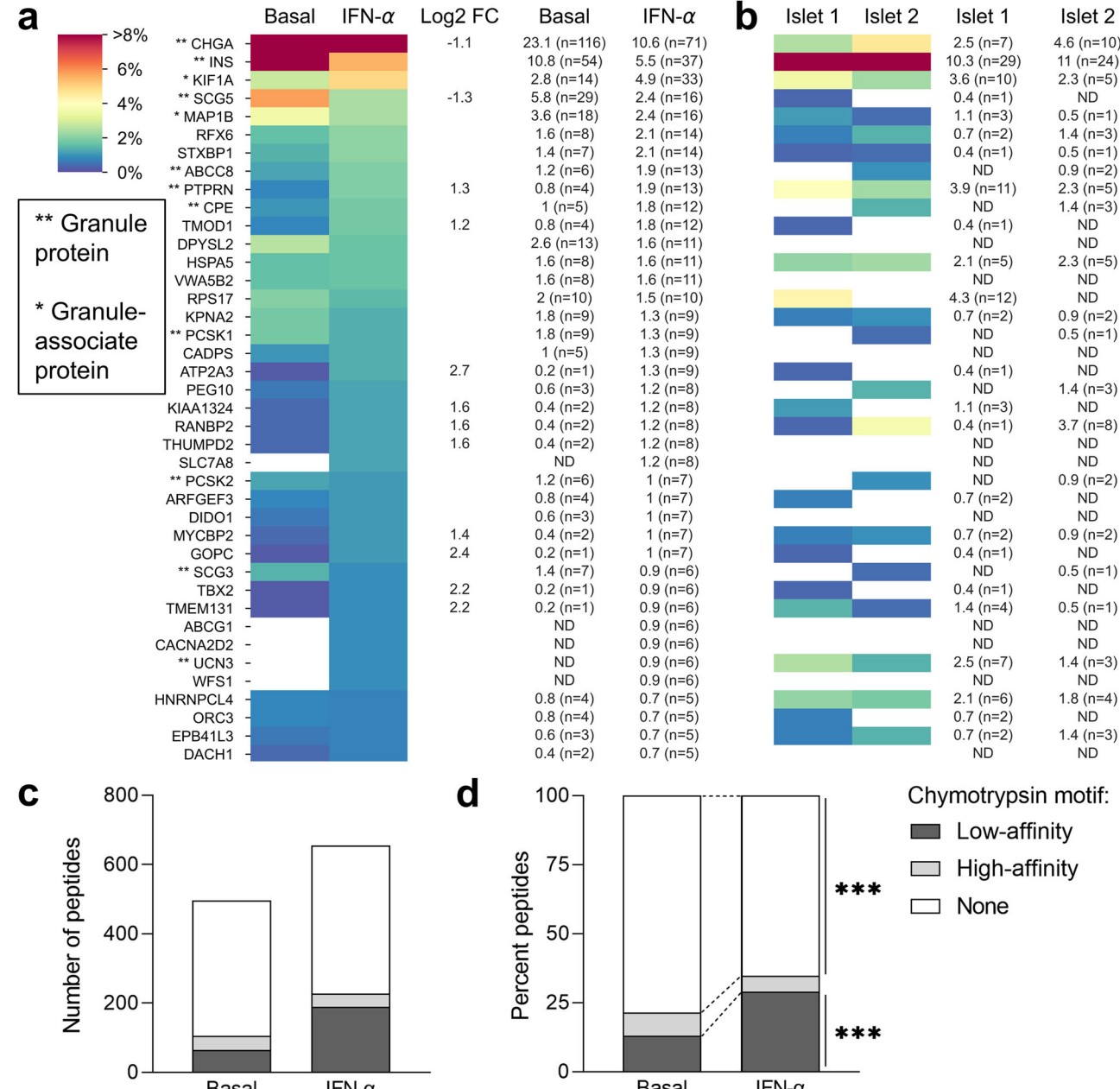

**Fig. 2 | β-cell immunopeptidome and chymotrypsin cleavage motifs.**
**a**, **b** Relative representation of the top 40 source proteins in ECN90 β-cells (**a**) and human islets (**b**), ranked according to the number of peptides detected in the IFN-α-treated condition. The color scale is proportional to the percent peptides (conventional and PTM) identified for each protein out of the total peptides in a given condition. PTM peptides were counted as such only for PTMs defined as likely biological (they were otherwise counted as unmodified); *cis*-spliced peptides were excluded. Percent values and peptide numbers are listed on the right. Source proteins enriched in IFN-α-treated cells (log₂ fold change, FC ≥ 1) or in basal condition (log₂ FC ≤ − 1) are indicated. The complete heatmap is provided in

Supplementary Fig. 2. HLA-I-bound peptides eluted from primary human islets are listed in Supplementary Data 5 and compared with those eluted from ECN90 β-cells in Supplementary Data 6. **c**, **d** Number (**c**) and percent (**d**) of HLA-I-eluted peptides across all 4 biological replicates carrying a chymotrypsin cleavage motif in basal and IFN-α-treated ECN90 β-cells. High-affinity cleavage motifs were defined as C-terminal Y, F, or W not followed by a P aa; low-affinity cleavage motifs were defined as C-terminal M or L not followed by a P aa. Additional analyses are reported in Supplementary Fig. 3 and Supplementary Data 7. ***$p < 0.0001$ by Fisher exact test.

peptides by characterizing immunopeptidome changes in ECN90 β-cells exposed or not to IFN-α and (immuno)proteasome inhibitors. Only the immuno-proteasome catalytic subunits were upregulated in IFN-α-treated ECN90 β-cells (Supplementary Fig. 3a). Accordingly, IFN-α treatment led to increased chymotrypsin-like activity, while treatment with carfilzomib, an inhibitor of the β5 proteasome subunit, or with ONX-0914, an inhibitor of the β5i immunoproteasome subunit with minimal cross-reactivity for its conventional β5 counterpart, decreased

such activity (Supplementary Fig. 3b). The HLA-I-eluted ligands were filtered and processed following the previous bioinformatics pipeline. The β-cell-enriched peptides retrieved across conditions clustered into distinct subgroups (Supplementary Fig. 3c and Supplementary Data 7). Some peptides (cluster 6) were specific to untreated β-cells and remained unaffected by constitutive proteasome inhibition. Conversely, peptides in cluster 3, also specific to untreated β-cells, were proteasome-dependent. Peptides in clusters 9 and 10 were instead largely specific to

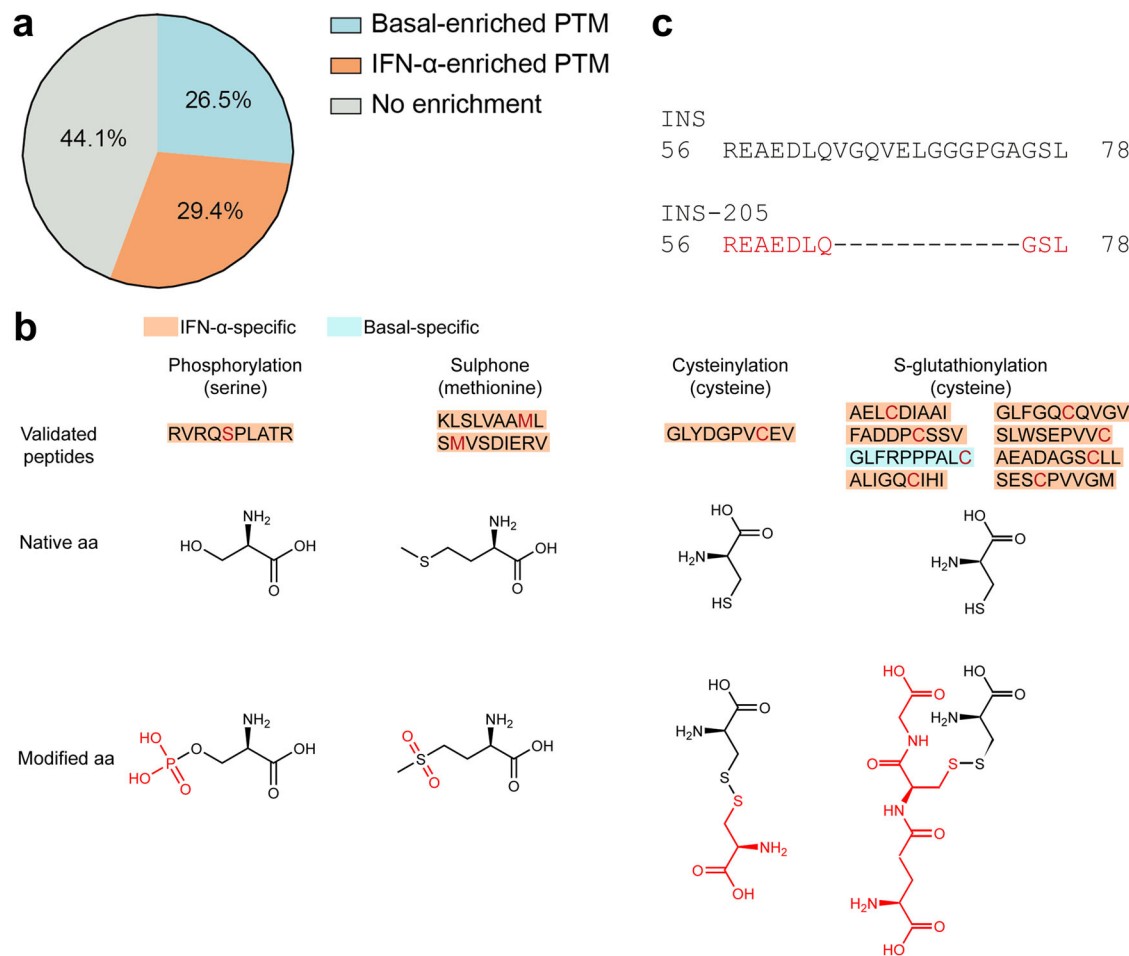

**Fig. 3 | Alternative antigen sequences. a** Global post-translational modification (PTM) enrichment in basal and IFN-α-treated conditions. A detailed list is provided in Supplementary Data 8. **b** Validated PTM peptides with representation of the native and modified aa (PTM in red). A detailed list is provided in Supplementary Data 3. **c** Peptide alignment of INS and INS-205 mRNA variant. The sequence of the HLA-eluted variant is indicated in red and corresponds to a sequence spliced out as compared to the canonical INS mRNA.

IFN-α-treated β-cells and, respectively, dependent and independent of immuno-proteasome catalysis. Of note, cluster 10 can be visually divided into peptides dependent (left half) or not (right half) on constitutive proteasome. Peptides in cluster 2 were dependent on constitutive or immuno-proteasome, regardless of IFN-α exposure. Similarly, the presentation of certain peptides was only enriched in β-cells treated with carfilzomib, regardless of IFN-α exposure (cluster 1). Peptides enriched upon immunoproteasome inhibition were also observed (cluster 8). Cluster 7, 4, and 5 were the most surprising. Cluster 7 depended on the exposure to either carfilzomib alone or a combination of IFN-α and carfilzomib/ONX-0914. Cluster 4 was enriched in β-cells either untreated or exposed to both IFN-α and ONX-0914, while cluster 5 was enriched in basal conditions, regardless of carfilzomib exposure, or β-cells treated with both IFN-α and ONX-0914. Of note, the preproINS (PPI)$_{15-24}$ peptide was not affected by (immuno)proteasome inhibition and was consistently identified in all conditions.

Collectively, these results show that the immunopeptidome is shaped by the chymotrypsin-like activity of both the constitutive and immunoproteasome expressed by β-cells.

**The immunopeptidome of ECN90 β-cells includes PTM, mRNA splicing and peptide *cis*-splicing alternative sequences enriched upon IFN-α treatment**

We subsequently focused on candidate alternative epitopes generated by PTMs and mRNA or peptide *cis*-splicing. Thirty-four PTMs were identified on HLA-I-eluted ligands from ECN90 β-cells (Supplementary

Data 8). Several of them (15/34, 44.1%) were found in the same proportion in both basal and IFN-α-treated samples (Fig. 3a). Since most PTM alternative epitopes are generated under cell stress conditions[17], we focused on those specifically enriched after IFN-α treatment (10/34, 29.4%). In order to not dismiss the opposite scenario of physiological PTMs that may be lost in an inflammatory environment, we also included those enriched under basal conditions (9/34, 26.5%). The second selection criterion was the possibility of chemically introducing the modification on synthetic peptides for MS validation, excluding PTMs that naturally arise during peptide synthesis or MS acquisition, e.g., oxidations and all PTMs found at the more vulnerable N- and C-terminal positions. Following these criteria, we retained for further study lysine acetylation, cysteinylation, deamidation, glutathione disulfide (S-glutathionylation), phosphorylation and sulphone and synthesized predicted HLA-A and -B binders in both modified and unmodified form. We then compared the spectrum of the identified PTM peptides: (a) with that of the synthetic PTM peptides; and (b) with the spectrum of the corresponding peptides synthesized in their native form (Supplementary Fig. 3d, e). In this second case, a spectral match would indicate an artifactual modification that is experimentally introduced. Only 12/31 *bona fide* biological PTM peptides were thus validated, carrying: phosphorylation (*n* = 1 peptide), sulphone (*n* = 2), cysteinylation (*n* = 1), and S-glutathionylation (*n* = 8) (Fig. 3b and Supplementary Datas 3, 8). Interestingly, 11/12 PTM peptides were exclusively found in IFN-α-treated cells. The immunopeptidome of human islets did not retrieve glutathionylated peptides.

Spectral matching of the 3 HLA-I-restricted mRNA splice candidates identified validated only one HLA-B40/B49-restricted INS-205 sequence (Fig. 3c), which was exclusively found in IFN-α-treated β-cells (Supplementary Data 1).

We further assigned 11 peptides which could hypothetically be explained by *cis*-splicing, either at the RNA or protein level. We could validate most spectra by matching them to the spectra of the synthetic counterpart (9/11, 81.8%), with some isoleucine/leucine ambiguities not resolvable by MS (Supplementary Data 4). A BamQuery analysis of these putative *cis*-spliced sequences did not retrieve any plausible alternative assignments to allegedly non-coding non-canonical mRNA reading frames[25,26]. Also, these *cis*-spliced peptides were mostly generated upon IFN-α treatment.

Altogether, 12 PTM peptides, 1 mRNA splice, and 9 candidate *cis*-spliced peptides were validated by spectral matching and found enriched upon IFN-α treatment.

## IFN-α skews peptide presentation toward HLA-B ligands

Another distinctive feature of IFN-α-treated ECN90 β-cells was the significant enrichment in HLA-B binders. In untreated cells, only 8% (41/502) of peptides were predicted HLA-B binders (-B40, -B49, or both), which increased to 34% (226/672) in IFN-α-treated cells (Fig. 4a; detailed in Supplementary Fig. 4). A similar but lesser effect was noted for HLA-C [5% (25/502) vs. 8% (56/672)], while the number of HLA-A and HLA-E ligands remained stable [41% (208/502) vs. 41% (277/672) and 0.8% (4/502) vs. 1.0% (7/672), respectively]. Thus, HLA-B ligands were the main contributor to the enriched peptide display induced by IFN-α. Using the current bio-informatics pipeline, we compared these findings with those of our previous datasets from ECN90 β-cells exposed or not to IFN-γ, alone or in combination with interleukin (IL)-1β[14]. Despite a 6.3-fold lower sequencing depth (Supplementary Fig. 5a), we identified a similar trend of preferential presentation of HLA-B ligands (Supplementary Fig. 5b).

When focusing on granule proteins, they contributed 32% of the HLA-I immunopeptidome in untreated ECN90 β-cells, which decreased to 22% upon IFN-α treatment (Fig. 4b), reflecting a dilutional effect due to the higher increase of peptides from other sources (Fig. 4c). Nonetheless, when analyzing HLA-A, -B and -C ligands separately, IFN-α increased the number of peptides from granule proteins only for HLA-B (Fig. 4c). The IFN-α-driven diversification of the immunopeptidome was also larger for HLA-B, which yielded 5.5-fold more peptides (226 vs. 41 in IFN-α-treated vs. untreated cells) compared to 1.3-fold (277 vs. 208) for HLA-A; and 4.2-fold (98 vs. 23) more source proteins compared to 1.2-fold (107 vs. 89) for HLA-A ligands (Fig. 4c and Supplementary Fig. 4). In terms of peptide abundance (Fig. 4d, e), the increase was also more evident for HLA-B ligands, particularly for granule-derived peptides, which did not significantly increase for HLA-A. The previously described HLA-A2-restricted preproINS (PPI)$_{15-24}$ peptide[14] was the most abundantly presented and was not affected by IFN-α treatment.

Using stringent criteria (i.e., no alternative restriction predicted), 7 conventional peptides predicted to bind HLA-E*01:01 were also identified, none of which was found exclusively in the basal condition, in line with the reported HLA-E upregulation induced by IFN-α[13]. A more extensive list of putative HLA-E ligands is provided in Supplementary Data 9.

Collectively, IFN-α skews antigen presentation toward HLA-B-restricted peptides.

## IFN-α preferentially upregulates HLA-B expression without inducing de-differentiation or reducing proINS synthesis

Based on these results, we hypothesized that the increased presentation of HLA-B ligands induced by IFN-α could reflect preferential HLA-B upregulation. Indeed, the median *HLA-B* gene expression increase was 31-fold in ECN90 β-cells treated with IFN-α, IFN-β, IFN-γ or

combinations thereof, while *HLA-A* was only marginally upregulated (median 6-fold) and *HLA-C* displayed an intermediate increase (median 19-fold; Fig. 5a). The overall effect of IFN-λ was more modest, while tumor necrosis factor (TNF)-α and IL-1β had no effect. We then validated HLA-A, -B, and -C Abs for their staining specificity, using HLA-I⁻ K562 cells transduced with different HLA-I alleles. Using flow cytometry (Supplementary Fig. 6a), HLA-A Ab ARC0588 and HLA-C Ab DT-9 only recognized their respective alleles, while HLA-B Ab JOAN-1 displayed some cross-reactivity with HLA-A alleles of low prevalence and a lesser reactivity to HLA-C variants. The HLA-A Ab ARC0588 was also functional and specific by western blot (Supplementary Fig. 6b), while the JOAN-1 and DT-9 Abs did not detect any band. Another Ab HC10, which was not functional by flow cytometry, preferentially recognized HLA-B (Supplementary Fig. 6b), with minimal cross-reactivity for HLA-C. Using these validated Abs, protein expression confirmed that, while only HLA-A was expressed under basal conditions, all IFNs preferentially upregulated HLA-B and, to a lesser extent, HLA-C, on ECN90 β-cells (Fig. 5b, c). Similar trends were noted for gene expression in non-dispersed primary human islets (Fig. 6a and Supplementary Table 1), and for surface protein expression in dispersed β-cells (Fig. 6b; gating strategy in Supplementary Fig. 6c), while preference was more biased toward HLA-C in α-cells. Differences did not always reach statistical significance, reflecting larger data dispersion (different islet preparations from different donors) and the higher basal HLA-I expression of islets compared to β-cell lines[14].

We then analyzed the effect of IFN-α on the expression of β-cell identity genes (Supplementary Fig. 7a). Despite a variable upregulation of the progenitor marker *SOX9*, several β-cell identity markers were also more consistently upregulated by IFN-α, i.e., *INS*, *CHGA*, *PCSK2*, *SYT4*, suggesting that de-differentiation was not induced. Using puromycin treatment (which is incorporated during protein synthesis) and an anti-puromycin Ab to detect newly synthesized proteins[27], IFN-α did not downregulate overall *de-novo* protein synthesis, while it upregulated HLA-I translation, as expected (Supplementary Fig. 7b, c). *De-novo* proINS synthesis was also unaffected (Supplementary Fig. 7d, e), at variance with the downregulating effect of IFN-γ on both total protein and proINS synthesis (Supplementary Fig. 7b, d, e).

Collectively, these results document a preferential HLA-B gene and protein upregulation by IFNs. IFN-α neither induced β-cell dedifferentiation nor downregulated proINS expression.

## Preferential HLA-B hyper-expression in the islets of T1D donors

These observations prompted us to test whether the histopathological hallmark of HLA-I hyper-expression in the islets of T1D patients[12] could also preferentially involve HLA-B. To this end, we stained pancreas tissue sections of T1D (n = 7) and non-diabetic donors (n = 6) from the Network for Pancreatic Organ Donors with Diabetes (nPOD) and Exeter Archival Diabetes Biobank (EADB; Supplementary Table 2), using the previously validated HLA-A Ab ARC0588, HLA-B Ab HC10 (the other HLA-B Ab JOAN-1 was not functional by immunofluorescence) and the HLA-A/B/C/E Ab EMR8-5[12] (Fig. 7a–c). As expected, an HLA-A/B/C/E hyper-expression was detected in INS-containing islets (ICIs) from T1D compared to non-diabetic (ND) donors, both in β-cells and α-cells (Fig. 7d and see Supplementary Fig. 8a–c for individual β-cells and α-cells of each T1D donor). However, HLA-B was hyper-expressed to a greater extent than HLA-A. Conversely, HLA-A/B/C/E, HLA-A and HLA-B expression in α-cells from T1D islets devoid of β-cells (INS-deficient islets, IDIs; Supplementary Fig. 8d) was lower than in α-cells from T1D ICIs and comparable to that of ND ICIs (Fig. 7d). Previous studies mapped the epitope recognized by the HLA-B Ab HC10[28], highlighting cross-reactivities with HLA-A allotypes A*25:01, A*26:01, A*30:01, A*31, A*33:03, A*34:01, A*66, A*68:02, A*69*01, A*80:01 and, possibly, A*10 and A*28. To exclude this potential confounder, we performed an additional analysis by excluding T1D case 6380 (A*33:03/A*68:02⁺) and

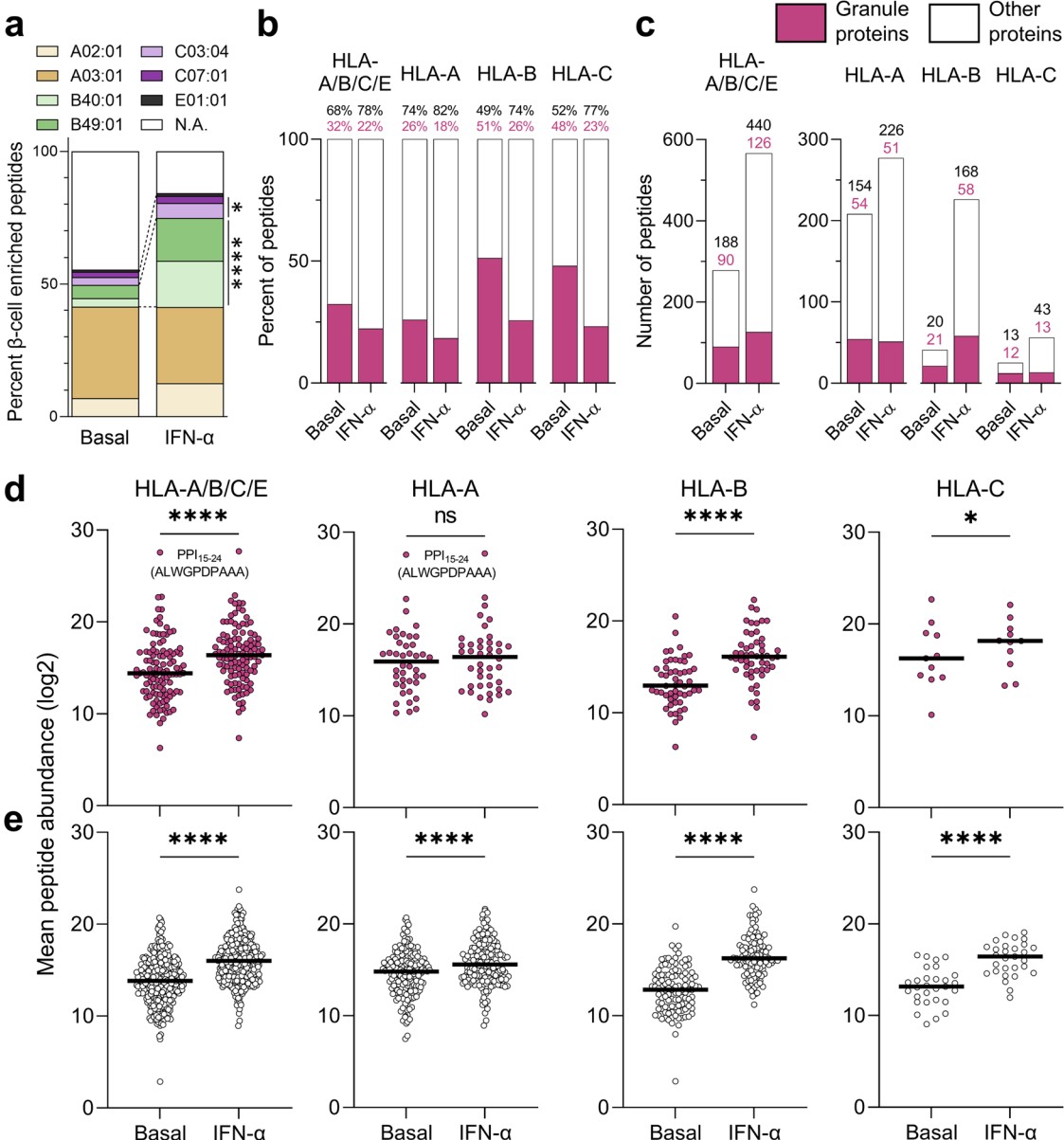

**Fig. 4 | HLA-I restrictions of the immunopeptidome of ECN90 β-cells exposed or not to IFN-α. a** Relative distribution of predicted HLA-I ligands for each allele expressed by ECN90 β-cells in basal and IFN-α-treated conditions across all 4 biological replicates. ****$p < 0.0001$ and *$p = 0.027$ by Fisher exact test. A heatmap of the source proteins of HLA-A- and HLA-B-restricted peptides is provided in Supplementary Fig. 4; additional analyses are presented in Supplementary Fig. 5. Predicted HLA-E*01:01-restricted peptides are listed in Supplementary Data 9. **b, c** Percent (**b**) and number (**c**) of peptides originating from granule-contained and non-diabetic cases 6278 (A*68:02+) and 6462 (A*25:01+) (Fig. 7d, crossed symbols), reaching similar conclusions.

other proteins in basal and IFN-α-treated conditions across all 4 biological replicates. **d, e** Mean total abundance of conventional peptides originating from granule-contained (**d**) and other proteins (**e**) across all 4 biological replicates. Peptides were identified by PEAKS and quantified by Progenesis. The PPI$_{15\text{-}24}$ sequence of the most abundant peptide identified in both conditions is indicated. Horizontal bars represent median values. ****$p < 0.0001$ and *$p = 0.042$ by Wilcoxon test.

Collectively, these results confirm the HLA-I hyper-expression in the islets of T1D individuals[12], and demonstrate a preferential hyper-expression of HLA-B over HLA-A.

### IFN-exposed β-cells preferentially enhance the activation and cytotoxicity of HLA-B-restricted CD8+ T cells

The next question was whether this preferential HLA-B upregulation translated into an enhanced activation of HLA-B-restricted CD8+ T cells against β-cells exposed to IFN-α. To this end, we first screened 101 T-cell receptors (TCRs), sequenced from islet-infiltrating CD8+ T cells of 4 HLA-B40+ T1D organ donors[29] from nPOD

(Supplementary Table 3) and re-expressed into ZsGreen-NFAT fluorescent reporter 5KC T cells[30], against 29 HLA-B40-restricted peptides derived from granule proteins and detected in IFN-α-treated β-cells (Supplementary Table 4). We identified a low-affinity TCR 173.D12 that recognized weakly but reproducibly a PPI$_{44\text{-}52}$ peptide (see below). We reasoned that the preferential HLA-B upregulation should translate into a preferential boosting of HLA-B-restricted T-cell activation upon exposure to IFN-treated β-cells. We, therefore, compared the ZsGreen-reported activation of HLA-B40-restricted PPI$_{44\text{-}52}$ 5KC T cells with that of HLA-A2-restricted PPI$_{15\text{-}24}$ counterparts upon exposure to ECN90 β-cells (Fig. 8a). HLA-A2-restricted T cells displayed higher activation than HLA-B40-restricted ones when exposed to untreated β-cells, which reflects

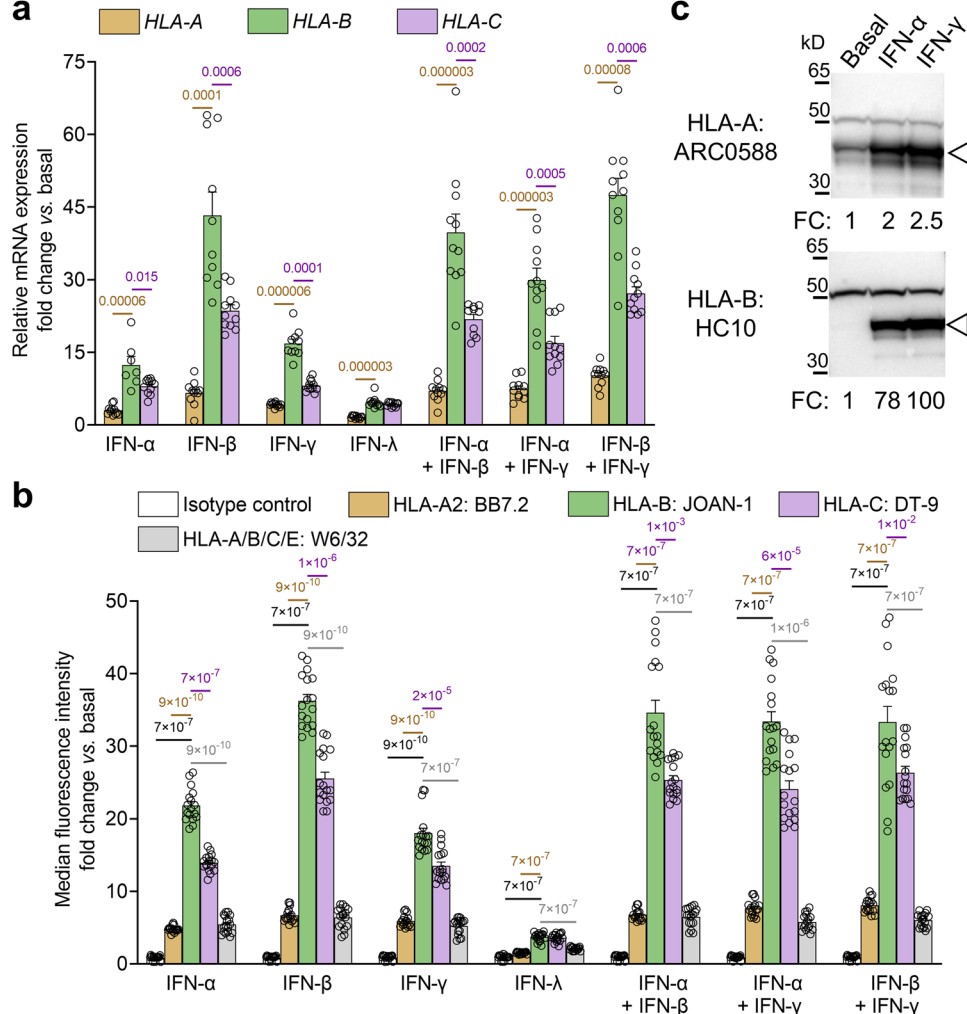

**Fig. 5 | IFNs preferentially upregulate HLA-B expression in ECN90 β-cells.**
**a** Relative mRNA expression (mean + SEM of 11 biological replicates) of *HLA-A*, *HLA-B*, and *HLA-C* genes in ECN90 β-cells exposed to the indicated IFNs for 24 h. *GAPDH* was used as an internal normalizing control, and each gene was normalized to the basal sample; *p*-values by Mann-Whitney U test indicated on top of bars. **b, c** Protein expression of HLA-A, -B and -C in ECN90 β-cells exposed to the indicated IFNs, as detected by surface flow cytometry (**b**; mean + SEM of 17 biological replicates) and

Western blot (**c**; representative experiment out of 3 performed) using the indicated Abs validated for their specificity (see Supplementary Fig. 6a, b). For flow cytometry, *p*-values by the Mann-Whitney U test are indicated on top of the bars. For Western blotting, the arrowhead indicates the HLA-I heavy chain band, with the top band indicating the α-tubulin loading control and normalized HLA-I fold change (FC) values indicated. Additional analyses are presented in Supplementary Fig. 7.

their higher affinity (Fig. 8b) and the higher abundance of $PPI_{15-24}$ presentation (Fig. 8c). However, HLA-A2-restricted T-cell activation was only marginally increased upon exposure to β-cells treated with IFN-α or IFN-γ, even when adding exogenous peptide to normalize antigen exposure. Conversely, HLA-B40-restricted T cells were poorly, if at all, stimulated by untreated β-cells, although some potentiation was observed with IFNs. When normalizing antigen presentation by $PPI_{44-52}$ peptide pulsing, the IFN-mediated enhancement was readily visualized (Fig. 8a). Both T cells exposed to *INS* knock-out (KO) ECN90 β-cells did not show any reactivity, irrespective of prior β-cell treatment, confirming their antigen specificity. Also, in this case, peptide pulsing enhanced HLA-B40-restricted T-cell activation to a larger extent.

To test whether the preferential HLA-B upregulation induced by IFN-α also translates into a superior enhancement of HLA-B-restricted cytotoxicity against β-cells, we transduced primary CD8+ T cells with an HLA-A2- or HLA-B40-restricted TCR reactive, respectively, to CMV and NY-ESO-1 non-β-cell peptides (Fig. 8b; $EC_{50}$ $3 \times 10^{-10}$ and $9 \times 10^{-6}$ nM, respectively), in order to normalize peptide presentation. Prior IFN-α exposure enhanced β-cell killing driven by HLA-B40-restricted, but not

HLA-A2-restricted CD8+ T cells (Fig. 8d; gating strategy in Supplementary Fig. 9a).

Collectively, these data indicate that IFN-exposed β-cells preferentially enhance the effector functions of HLA-B-restricted CD8+ T cells.

## Islet-infiltrating CD8+ T cells from HLA-B40+ T1D patients recognize HLA-B40-restricted β-cell peptides

Last, we asked whether this preferential HLA-B upregulation in insulitis lesions translated into recognition of HLA-B-restricted peptides by islet-infiltrating T cells. To this end, we analyzed HLA-B40-restricted responses in CD8+ T-cell lines raised from islet infiltrates of 4 HLA-B40+ nPOD T1D donors[31] (Supplementary Table 3). The previous 29 HLA-B40-restricted peptides were first screened by combining them into 4 pools (Supplementary Table 4), using peptides binding to irrelevant HLA-I alleles not expressed by donors as negative controls (Supplementary Table 5). All 4 donors yielded some T-cell lines with positive IFN-γ responses to most peptide pools (Fig. 9a–c and Supplementary Fig. 9b). Overall, reactivities were found in 19/19 positive T-cell lines for peptide pool 1, 16/19 for pool 2, 15/19 for pool 3, and 18/19 for pool 4.

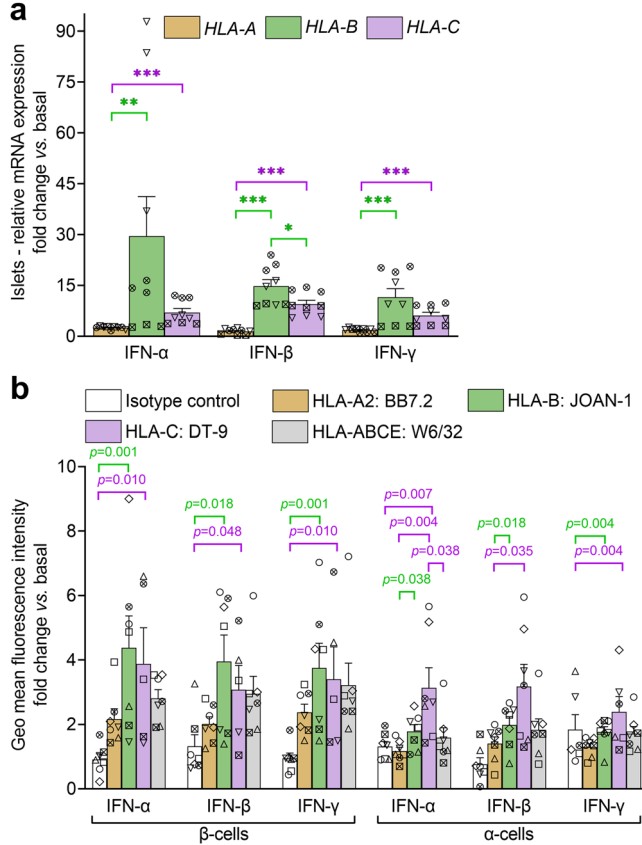

**Fig. 6 | IFNs preferentially upregulate HLA-B expression in human islets.**
**a** Relative mRNA expression (mean + SEM) of *HLA-A*, *HLA-B*, and *HLA-C* genes in primary human islets exposed to the indicated IFNs for 24 h. Data normalization is the same as in Fig. 5a, and each symbol represents triplicate measurements from islets of a single donor (Supplementary Table 1); ***$p$ = 0.00004, **$p$ = 0.0005, *$p$ = 0.014 by Mann-Whitney U test. **b** Surface expression (mean + SEM of 7 biological replicates; gating strategy in Supplementary Fig. 6c) of HLA-A, -B and -C in β-cells and α-cells from primary human islets of 7 non-diabetic organ donors (Supplementary Table 1). Each symbol represents one organ donor; *p*-values by the Mann-Whitney *U* test.

The reactivity of 3 positive CD8+ T-cell lines (Fig. 9a–c) was deconvoluted by testing for individual peptides from positive pools (Fig. 9d–f). Significant IFN-γ secretion was observed in 2 independent T-cell lines for peptides ABCC8$_{1218-1226}$ and ABCC8$_{1557-1565}$; CHGA$_{268-275}$ and CHGA$_{323-332}$; CPE$_{203-211}$ and CPE$_{358-366}$; and C$_{926}$-glutathionylated KIF1A$_{923-931}$. Strong responses in single T-cell lines were also measured for peptides CHGA$_{442-451}$, INS$_{66-77}$, SCG5$_{138-146}$ and SCG5$_{142-150}$.

Collectively, these results show that multiple HLA-B*40-restricted peptides are targeted by islet-infiltrating T cells from HLA-B40+ T1D donors.

## Discussion

Our study provides an in-depth view of the repertoire of HLA-I-bound peptides presented by β-cells exposed or not to the early T1D signature cytokine IFN-α. Expectedly, IFN-α exposure increased HLA-I surface expression[2,13], which resulted in a higher number and abundance of presented peptides, as we previously reported for IFN-γ[14]. Thus, the inflammatory milieu of insulitis likely enhances the antigenic visibility of β-cells, both qualitatively and quantitatively. These peptides originated from a wider array of source proteins, possibly reflecting IFN-α-induced ER stress[2] and increased numbers of misfolded proteins undergoing proteasomal degradation[17]. While the peptide fraction derived from secretory granule proteins decreased due to a dilutional

effect, both the number and abundance of granule-derived peptides increased, underlining the major contribution of secretory granules to the pHLA-I display of β-cells[14,32] under basal and, to a larger extent, IFN-α-treated conditions. In line with previous findings[33], de-differentiation, which could have resulted in increased degradation of granule proteins and presentation of derived peptides, was not at play, as IFN-α rather increased the expression of β-cell identity genes. Accordingly, proINS synthesis was also unaffected by IFN-α, while it was decreased by IFN-γ. β-cell de-differentiation might thus be a late event in T1D pathogenesis, induced by inflammatory cytokines that intervene later than IFN-α[34] and may represent a defense mechanism to limit auto-immune vulnerability[35]. At earlier disease stages, the HLA-E upregulation induced by IFN-α[13] may provide another line of defense. HLA-E inhibits NK-cell-mediated cytotoxicity and presents a limited set of peptides to regulatory CD8+ T cells[36]. Accordingly, the peptides predicted to bind exclusively to HLA-E were presented only by IFN-α-treated β-cells and will provide a useful resource for follow-up studies.

The immunopeptidome obtained confirmed our previous findings[14] that some known β-cell antigens are underrepresented (ZnT8, GAD65, IAPP; with IAPP possibly reflecting a model bias, as ECN90 express low IAPP levels[20]) or even absent (IGRP, INS DRiP), despite a much higher sequencing depth yielding 6.3-fold more peptides. This does not exclude that these missing antigens may become T-cell targets upon peptide presentation by antigen-presenting cells phagocytosing β-cell material. The INS DRiP case is noteworthy, as it has been confirmed as a relevant T-cell antigen targeted by islet-infiltrating CD8+ T cells[29]. The short-lived and unstable nature of INS DRiP and the fact that its presentation is enhanced by IFN-γ but not IFN-α[37] may explain this discrepancy, emphasizing the need for complementary antigen identification strategies[38]. The proteasome-independent generation of some peptides identified may reflect their processing in other cell compartments. One notable example is PPI$_{15-24}$, which is reportedly processed by signal peptidases during the import of nascent proINS into the ER[17]. We also identified novel antigens, namely CHGB, SCG2, and SCG3, which share several features with the previously reported SCG5, UCN3, and PCSK2[14,32]. They are all soluble granule proteins undergoing intermediate processing by proconvertases and furins to yield their bioactive products, which are released with secretory granules. This processing, along with the modulation of their biosynthesis according to metabolic demand and inflammatory context, may easily divert them toward the HLA-I pathway under β-cell stress conditions[17].

The comparison of the immunopeptidome of ECN90 β-cells and primary human islets was also informative. Several peptides restricted for the shared HLA-A2/A3 alleles identified in ECN90 β-cells were also found in islets, lending support to this model. Intriguingly, several GCG-derived peptides were also retrieved from islets (*n* = 43 vs. 33 for INS) (Supplementary Fig. 10). Interestingly, α-cells displayed an HLA-I hyper-expression equivalent to that of β-cells in T1D donors compared to non-diabetic controls, but less so upon short-term in vitro exposure to IFNs, suggesting that prolonged inflammation is required to upregulate HLA-I. Moreover, the IFN-induced upregulation was more biased toward HLA-C in α-cells. These findings suggest that the autoimmune resistance of α-cells cannot be ascribed to a lesser antigen presentation leading to immune ignorance and that this presentation may be differently distributed across HLA restrictions. Other α-cell-intrinsic defense mechanisms and/or a higher proneness of the T-cell repertoire to recognize INS rather than GCG[29,39] may be at play. Moreover, the *GCG* gene is more expressed than *INS* in mature human mTECs[36,40].

Our thorough investigation of HLA-eluted alternative peptide sequences led to the robust identification of 1 HLA-B40/B49-restricted INS-205 mRNA splice peptide, 9 *cis*-spliced candidates, and 12 sequences bearing PTMs. Among these PTMs, citrullination, a relevant PTM in T1D[41,42], was not found, including after additional spectral

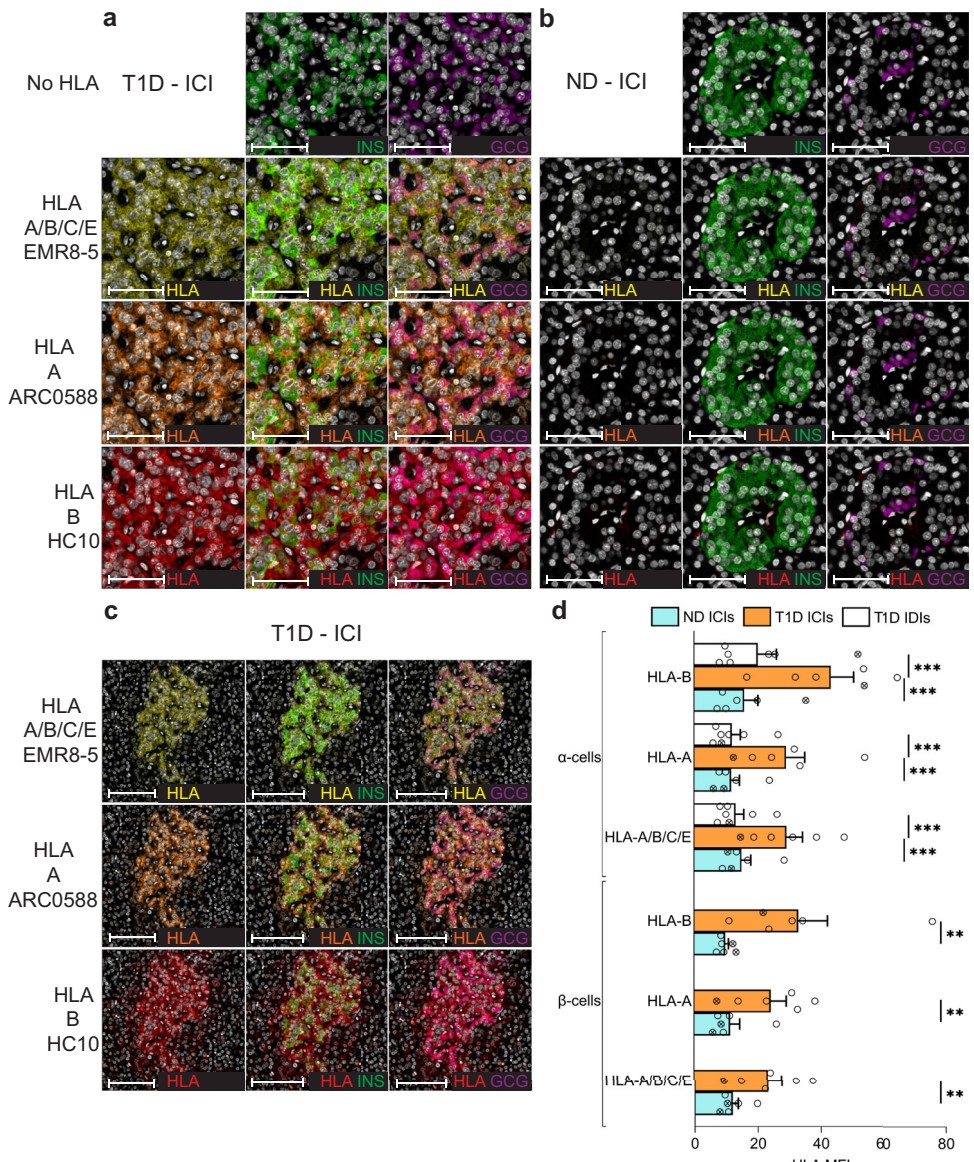

**Fig. 7 | HLA-B vs. HLA-A hyper-expression in the islets of T1D and non-diabetic (ND) cases. a, b** Representative immunofluorescence images of insulin-containing islets (ICIs) from T1D case nPOD 6396 (**a**) and ND case nPOD 6160 (**b**; all cases listed in Supplementary Table 2), stained with DAPI only (first row) or for HLA-A/B/C/E (yellow; second row), HLA-A (orange; third row) and HLA-B (red, fourth row), alone (first column) or in combination with INS (green, second column) or GCG (violet, third column). Scale bar 50 μm. **c** Whole ICI images from the same T1D case nPOD 6396, scale bar 100 μm. **d** Immunofluorescence quantification of HLA-I mean fluorescence intensity (MFI; mean + SEM) for HLA-A/B/C/E, HLA-A and HLA-B in β-cells (bottom) and α-cells (top) from ICIs of ND (blue; $n = 6$) and T1D cases (orange; $n = 6$); and in α-cells from insulin-deficient islets (IDIs) of T1D cases (white; $n = 7$). Crossed symbols indicate ND cases nPOD 6278 and 6462 and T1D case nPOD 6380 harboring HLA-A allotypes potentially cross-reactive with the HLA-B Ab HC10; **$p = 0.01$ and ***$p ≤ 0.0002$ by 2-way ANOVA after exclusion of these 3 cases. T1D IDI images and individual quantifications for each T1D donor are provided in Supplementary Fig. 8.

matching analysis to exclude misidentification as glutamine- or asparagine-deaminated peptides[43]. It remains possible that citrullinated epitopes may be generated outside β-cells. The most represented PTM (8/12 peptides) was S-glutathionylation, which involves the formation of a disulfide bond between a cysteine residue and a glutathione tripeptide. Glutathione is a key circulating anti-oxidant, whose levels are reduced in T1D patients due to consumption[44]. It participates in INS degradation by reducing the disulfide bonds between the A and B chain[45]. The observation that more HLA-I-bound glutathionylated peptides are found in IFN-α-treated conditions may reflect the altered redox balance of inflamed β-cells[46], a promising therapeutic target in recent trials[47]. Moreover, epitope glutathionylation can alter T-cell recognition[48], implying that oxidative stress may also contribute to the antigenic visibility of β-cells. Cysteines are

exquisitely sensitive to oxidative PTMs, with cysteinylation (i.e., a disulfide-linked cysteine addition) also found here and previously reported in other fields[49,50]. Although unavoidable experimental artifacts preclude a comprehensive assessment of the PTM peptidome, most of the stringently validated PTM peptides (11/12), cis-spliced peptides (6/9), and the INS-205 mRNA-spliced peptide were found exclusively after IFN-α treatment, underlining the role of IFN-α in shaping this alternative antigen landscape. This alternative antigen generation may further increase the autoimmune vulnerability of β-cells.

Besides HLA-I hyper-expression and alternative antigen generation, preferential HLA-B upregulation by IFN-α may contribute to the increased antigenic visibility of β-cells. This effect is not unique to β-cells or IFN-α, as it has been reported for IFN-γ in cancer cells[15,51]. HLA-B

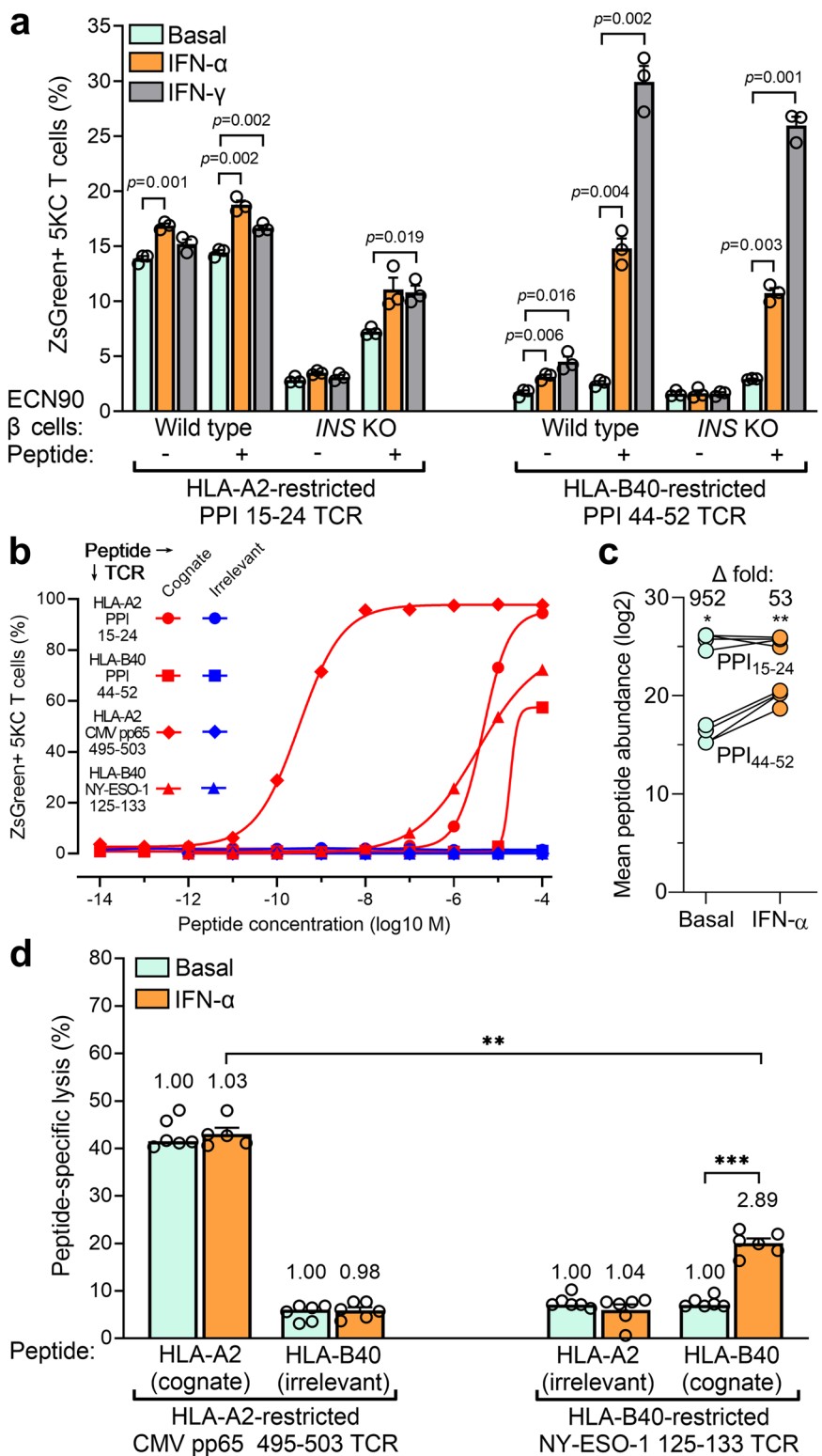

expression is also preferentially silenced in tumors escaping immune recognition[52,53], and enriched in extracellular vesicles[54]. Preferential HLA-B upregulation translated into an increased peptide presentation drastically skewed toward HLA-B ligands. Although remaining unnoticed, a preferential upregulation of HLA-B and, to a lesser degree, HLA-C was also observed in RNA-seq and proteomics studies of islets, stem cell-derived islet-like cells, and β-cell lines exposed to IFNs[16,55–57], and of sorted β-cells from T1D vs. non-diabetic donors[46,58,59]. Mechanistically, this may be driven by the *HLA-B* locus-specific IFN response element, which has a higher affinity for IFN regulatory factor-1 and -2[60,61]. Ubiquitin-mediated degradation of HLA-A but not HLA-B mRNA may also be at play[62]. Accordingly, the activation and cytotoxicity of HLA-B40-restricted T cells were boosted when β-cells were exposed to IFNs, while those of HLA-A2-restricted T cells were only marginally increased. Moreover, islet-infiltrating CD8+ T cells from T1D patients recognized HLA-B40-restricted granule peptides, notably derived from the known β-cell antigens INS, CHGA, SCG5, the newly described ATP-binding cassette subfamily C member 8 (ABCC8) and

**Fig. 8 | Enhanced activation of HLA-B-restricted CD8⁺ T cells by IFN-treated β-cells. a** Activation of ZsGreen-NFAT reporter 5KC T cells transduced with a 1E6 TCR recognizing HLA-A2-restricted PPI$_{15-24}$ or a 173.D12 TCR recognizing HLA-B40-restricted PPI$_{44-52}$. Following the indicated IFN pretreatment, ECN90 β-cells (wild-type or *INS* knock-out, KO) left unpulsed or pulsed with the TCR-cognate peptide were co-cultured with TCR-transduced 5KC T cells for 6 h. Data represent mean + SEM of triplicate measurements from a representative experiment out of 3 performed; *p*-values by Student's *t* test. **b** Dose-response peptide recall of reporter 5KC T cells reactive to HLA-A2/PPI$_{15-24}$, HLA-B40/PPI$_{44-52}$ (used in panel **a**); HLA-A2/CMV pp65$_{495-503}$, HLA-B40/NY-ESO−1$_{125-133}$ (TCRs used in panel **d**). 5KC T cells were co-cultured for 18 h with K562 antigen-presenting cells transduced with HLA-A2 or HLA-B40 and pulsed with the indicated peptides. A representative experiment out of 2 performed is shown. **c** Average total abundance and fold difference (Δ) of HLA-

A2-restricted PPI$_{15-24}$ and HLA-B40-restricted PPI$_{44-52}$ peptides presented under basal and IFN-α-treated conditions (*n* = 4/each). *\*p* = 0.016 and *\*\*p* = 0.004 by paired Student's *t* test. **d** Cytotoxic lysis of ECN90 β-cells pulsed with HLA-A2-restricted CMV or HLA-B40-restricted NY-ESO-1 peptide. HLA-A2/CMV- or HLA-B40/NY-ESO-1-reactive TCR-transduced CD8⁺ T cells were cultured for 18 h with a fixed mixture of peptide-pulsed CFSE-labeled β-cells and unpulsed CTV-labeled β-cells, both preliminary treated or not with IFN-α for 24 h. The gating strategy is presented in Supplementary Fig. 9a. Percent-peptide-specific lysis is indicated, as calculated from the ratio of surviving (Live/Dead⁻) pulsed CFSE⁺ vs. unpulsed CTV⁺ β-cells normalized to the same ratio in wells containing β-cell targets alone. Data represents mean + SEM of sextuplicate measurements from a representative experiment out of 2 performed, with fold-increase lysis vs. the basal condition indicated on top of each bar. *\*\*\*p* = 0.005 and *\*\*p* = 0.008 by Mann-Whitney *U* test.

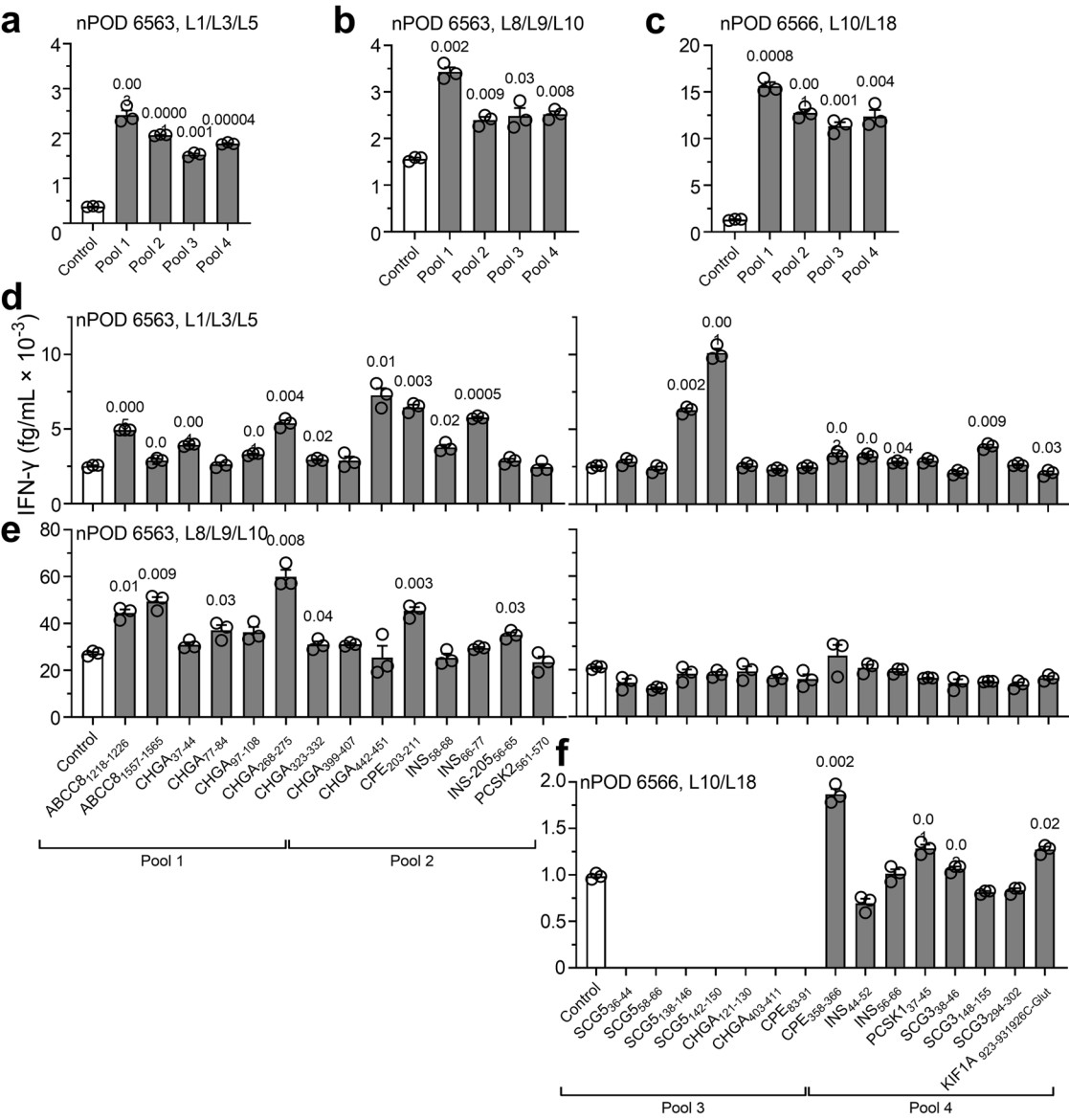

**Fig. 9 | HLA-B40-restricted peptides recognized by islet-infiltrating T cells. a–c** IFN-γ secretion by polyclonal CD8⁺ T-cell lines expanded from islet infiltrates of HLA-B40⁺ nPOD T1D donors (listed in Supplementary Table 3) and exposed to HLA-B40-transduced K562 antigen-presenting cells pulsed with HLA-B40-restricted peptide pools (peptides listed in Supplementary Table 4). **d–f** The same CD8⁺ T-cell lines exposed to HLA-B40-transduced K562 antigen-presenting cells pulsed with

individual HLA-B40-restricted peptides from the previous pools. Data represent mean + SEM of triplicate measurements from a representative experiment out of 2 performed; *p*-values by paired Student's *t* test indicated on top of bars. Additional T-cell lines responding to HLA-B40-restricted peptide pools are presented in Supplementary Fig. 9b.

carboxypeptidase E (CPE), and glutathionylated KIF1A. Given the different peptide-binding preferences of each HLA-I type, this implies that the inflammatory microenvironment of insulitis may skew the autoimmune response toward a distinct set of HLA-B-restricted peptides and recruit additional CD8[+] T-cell clonotypes. Under physiological conditions, HLA-B-restricted peptides may contribute very marginally to the β-cell antigen display (immune ignorance). Conversely, in the local inflammatory islet microenvironment of T1D (insulitis), type I IFNs may drive the preferential exposure of these HLA-B-restricted peptides and thus contribute to tolerance breakdown through loss of immune ignorance and epitope spreading (Supplementary Fig. 11). This finding is also relevant in view of the paucity of HLA-B-restricted β-cell epitopes described[19]; and of the notion that HLA-B*39:06 is the strongest T1D-predisposing HLA-I allele (relative risk 5.6)[63]. Moreover, it has been reported that HLA-B but not HLA-A matching improves islet[64] and kidney[65] allograft survival. This paradigm of preferential HLA-B upregulation may also apply to other autoimmune diseases[66] and cancers[15] featuring IFN signatures.

Besides the higher peptide yield, our current work significantly enriches our previous analysis of the β-cell immunopeptidome[14]. First, the issue of alternative antigens, which was only marginally addressed in our previous work, is here developed to identify PTMs, and *cis*-spliced peptides using the dedicated MARS algorithm[21]. Second, the present work addresses another biological question, which is the increased immune visibility of β-cells induced by IFN-α and other IFNs. This differential β-cell immunopeptidome between basal and IFN-treated conditions was also marginally analyzed in our previous work, due to more limited quantification tools and lower peptide yields. Third, this deeper peptide identification and improved, fully automated bio-informatics pipeline relying on de novo sequencing highlighted the preferential presentation of HLA-B ligands that previously went unnoticed. Re-analysis of the previous datasets with this new pipeline revealed a similar trend for IFN-γ, in line with the preferential HLA-B upregulation induced by all IFNs.

This study also carries limitations. First, our work focused on predicted HLA-I ligands, and the datasets used to train prediction algorithms are mostly derived from viral and tumor immunology studies. While in these studies HLA-I binding is a good predictor of T-cell immunogenicity, it likely underestimates the number of potential epitopes in the case of autoimmunity[38], possibly reflecting a bias imposed by thymic T-cell deletion. Predicted non-binders may hide additional immunogenic peptides[38]. Second, we analyzed T-cell recognition in only a subset of HLA-B40-restricted peptides. The large repertoire of candidate epitopes identified will require higher-throughput technologies based on DNA-barcoded HLA-I multimers[67] for comprehensive validation. Third, further research is needed to conclusively demonstrate that the increased presentation of HLA-B-restricted endogenous peptides leads to increased β-cell killing.

In conclusion, our study shows that IFN-α shapes the immunopeptidome of β-cells, promoting alternative antigen formation, increasing the presentation of HLA-B-restricted peptides, and enhancing the cytotoxicity of HLA-B-restricted CD8[+] T cells against β-cells. Islet-infiltrating CD8[+] T cells recognize a subset of these peptides derived from granule proteins. This comprehensive catalog of HLA-I-eluted peptides sheds light on a neglected, distinct set of peptides presented by HLA-B molecules and invites further studies to identify diabetogenic HLA-B-restricted CD8[+] T cells and T-cell biomarkers.

## Methods
### Ethics statement
All recruited human subjects or next of kin provided written informed consent, without participant compensation. Studies were approved by the relevant Ethics Committees indicated in each Methods section.

### β-cell culture and treatments
The human ECN90 β-cell line[20] (Human Cell Design) was derived from the neonatal pancreas of a 4-month-old patient suffering from hyperinsulinemic hypoglycemia in infancy. It was maintained in DMEM/F12 Advanced medium (ThermoFisher) supplemented with 2% bovine serum albumin (BSA; fraction V, fatty-acid-free; Roche), 50 μM β-mercaptoethanol (Sigma), 10 mM nicotinamide (Merck), 1.7 ng/mL sodium selenite (Sigma), 100 μ/mL penicillin-streptomycin (Gibco), 1% GlutaMAX (Gibco). Cells were seeded at 8-8.5 × 10⁶ cells in 75 cm² or 16.5 × 10⁶ cells in 150 cm² culture flasks (TPP) coated with 0.25% fibronectin from human plasma (Sigma) and 1% extracellular matrix from Engelbreth-Holm-Swarm murine sarcoma (Sigma) and cultured for 18–24 h. ECN90 cells were treated in DMEM/F12 medium (ThermoFisher), supplemented as above or without BSA with the following agents: IFN-α (PBL Assay Science, #11100-1; 2000 μ/mL), IFN-γ (R&D, #285-IF-100; 500 μ/ml), IFN-β (R&D, #8499-IF; 2000 μ/mL), IFN-λ1 (PeproTech, #300-02 L; 286 μ/mL), TNF-α (R&D, #210-TA; 1000 μ/mL), IL-1β (R&D, #201-LB; 1000 μ/mL), carfilzomib (Selleckchem, #S2853; 50 nM) or ONX-0914 (Selleckchem, #S7172; 100 nM). The proteasome enzymatic activity was measured using Proteasome-Glo (Promega, #G1180).

For immunopeptidomics experiments, human pancreatic islets were obtained from 2 non-diabetic brain-dead organ donors (79-year-old male, BMI 27.8 kg/m², insulin secretion 25.2 and 51.0 μU/ml at 3.3 and 16.7 mM glucose, respectively; 80-year-old female, BMI 21.6 kg/m², insulin secretion 19.6 and 29.6 μU/ml at 3.3 and 16.7 mM glucose, respectively); protocol approved by the Ethics Committee of the University of Pisa, Italy.

To evaluate HLA-I expression, human islets were obtained from the Integrated Islet Distribution Program (IIDP; Supplementary Table 1) and maintained in CMRL1066 medium. Upon receipt, islets were left to recover for 3 h before adding or not IFN-α, IFN-β, or IFN-γ as above.

The *INS* KO ECN90 β-cell line was generated by transfection with Lipofectamine CRISPRMAX (Invitrogen, #CMAX00001) with Alt-R S.p. Cas9 Nuclease V3 (IDT, #1081058) and *INS*-targeting gRNA (IDT). *INS* KO was validated by real-time quantitative (RT-q)PCR and by INS ELISA (Mercodia, #10-1113-01).

### HLA immunoprecipitation and peptide elution
Purified anti-HLA-A/B/C/E Ab W6/32 (8–16 mg; produced in-house) was incubated with protein A Sepharose beads for 30 min before washing with borate buffer (0.05 M boric acid, 0.05 M KCl, 4 mM NaOH, pH 8.0). Bound Abs were cross-linked to the beads with 40 mM dimethyl pimelimidate dihydrochloride (Sigma) in borate buffer pH 8.3 for 30 min. Cross-linking was terminated with ice-cold 0.2 M Tris pH 8.0, and unbound Ab was removed by washing with 0.1 M citrate buffer pH 3.0 followed by 50 mM Tris pH 8.0.

Dry-frozen cell pellets (1 × 10⁸) were lysed in 1 mL lysis buffer (1% IGEPAL-CA 630, 300 mM NaCl, 100 mM Tris pH 8.0, 1X Roche cOmplete Mini Protease Inhibitor Cocktail, EDTA-free) by mixing at 4 °C for 30 min. Human primary islets were lysed in lysis buffer in bead beater tubes (Precellys Evolution, Bertin) following 5 cycles at 7200 rpm for 20 s, separated by 20 s pauses. Lysates were collected, further mixed for 20 min at 4 °C, and cleared by centrifugation at 500 × *g* for 10 min followed by 21,000 × *g* for 45 min at 4 °C. pHLA complexes were captured by incubating the cleared lysates with W6/32 Ab-cross-linked beads overnight at 4 °C. The resin was collected by gravity and subjected to 4 washes in Tris buffer 50 mM, pH 8.0 containing 1) 150 mM NaCl, 5 mM EDTA; 2) 150 mM NaCl; 3) 450 mM NaCl; 4) no addition. Peptides were eluted with 10% acetic acid.

### HPLC fractionation and purification of HLA-I-bound peptides
ECN90 samples were resuspended in 120 μL loading buffer (0.1% TFA, 1% acetonitrile) and injected in an Ultimate 3000 HPLC System (ThermoFisher). Peptides were separated across a 4.6- by 50 mm

ProSwift RP1S column (ThermoFisher) using 1 mL/min flow rate over 10 min from 2% to 35% buffer B (0.1% TFA in acetonitrile) in buffer A (0.1 TFA in water). Fifteen fractions were collected every 30 s, and peptide fractions 1–9 were combined into odd and even aliquots to obtain 2 technical replicates per sample, then vacuum-dried prior to LC-MS/MS acquisition.

Resuspended HLA-I-eluted samples were centrifuged through 5 kDa cutoff filters (Merck/Millipore #UFC3LCCNB-HMT) and vacuum-dried. They were then resuspended in a loading buffer and cleared using Pierce C18 Spin Tips (ThermoFisher, #84850). Final elution was performed in 30% acetonitrile 0.1% TFA. Samples were vacuum-dried prior to LC-MS/MS acquisition.

## LC-MS/MS acquisition

Dried samples of HLA-I-bound peptides from ECN90 β-cells (4 biological replicates per condition, i.e., with or without IFN-α; 2 technical replicates per sample, i.e., odd and even HPLC fractions) and synthetic peptides were resuspended in loading buffer and acquired on an Ultimate 3000 RSLCnano system coupled to a Fusion Lumos mass spectrometer (ThermoFisher). The 2 technical replicates of each of the 4 biological replicates were pooled for subsequent data analysis. Peptides were separated using a PepMap C18 column, 75 μm × 50 cm, 2 μm particle size (ThermoFisher) with a 30 min (synthetic peptides) to 60 min linear acetonitrile in water gradient of 2-25%, at a flow rate of 250 μL/min. The solvent contained 5% DMSO and 0.1% formic acid (v/v). Peptides were ionized using an EasySpray source at 2000 V and ions were introduced into the mass spectrometer through an on-transfer tube at 305 °C. The data-dependent acquisition was performed with one full MS1 spectra recorded from 300 to 1500 m/z (120,000 resolution, 400,000 AGC target, 60 ms accumulation time), followed by MS2 scans (30,000 resolution, 120 ms accumulation, 300,000 AGC target). Precursor selection was performed using Top-Speed mode at a cycle time of 2 s. High-collision dissociation fragmentation was induced at an energy setting of 32 for singly charged peptides and 28 for peptides with a charge state of 2–4.

Alternatively, the 15 vacuum-dried samples of HLA-I-bound peptides from ECN90 β-cells exposed to (immuno-)proteasome inhibitors (i.e., 5 conditions, 3 biological replicates/each) or the 2 samples of human islets were resuspended in SCP loading buffer (1% acetonitrile, 0.1% formic acid) and analyzed by a nanoElute system coupled to a timsTOF SCP mass spectrometer (Bruker). Peptides were loaded into an Aurora C18 column, 25 cm × 75 μm, 1.7 μm particle size (IonOpticks) with a loading pressure of 800 bars for 9 min and were separated with a linear acetonitrile gradient of 2–25% in 0.1% acetic acid over 60 min and 24–37% for additional 6 min at a flow rate of 150 nL/min at 50 °C. Peptides were ionized with the CaptiveSpray source (Bruker) at 1400 V and 180 °C. Data were acquired in DDA PASEF mode with one TIMS-MS survey and 10 PASEF MS2 scans per cycle. Ion accumulation and ramp time in the TIMS analyzer were set to 166 ms/each. The ion mobility range for peptide analysis was set to $1/K0 = 1.7$ to $0.7$ Vs/cm$^2$, and the m/z range was 100–1700. Two compound regions were defined using m/z and $1/K0$ Vs/cm$^2$ as follows: 300, 0.7; 800, 1.2; 800, 0.9; 500, 0.7; and 700, 1.4; 700, 1.1; 1000, 1.7; 1500, 1.7. Precursors with charge states 1–3 and a minimum threshold of 500 arbitrary units were fragmented and re-sequenced until reaching a "target value" of 20,000 arbitrary units. Collision energies were 70 eV at $1/K0 = 1.7$ Vs/cm$^2$; 40 eV at $1/K0 = 1.34$ Vs/cm$^2$; 40 eV at $1/K0 = 1.1$ Vs/cm$^2$; 30 eV at $1/K0 = 1.06$ Vs/cm$^2$; 20 eV at $1/K0 = 0.7$ Vs/cm$^2$.

## LC-MS/MS data analysis

Raw data analysis was performed using PEAKS X or PEAKS X Pro (Bioinformatics Solutions), with a precursor mass tolerance of 10 ppm and a fragment ion mass tolerance of 0.03 Da (for samples run on timsTOF SCP) or 0.05 Da (for samples run on Fusion Lumos), setting the enzyme to "none" and the digestion to "unspecific", reporting 5–10 peptides for each spectrum in *de-novo* sequencing. The search was performed on a protein sequence FASTA file containing the reviewed human UniProt entries (Swiss-Prot, downloaded on 22/01/2019), predicted alternative peptide sequences translated from published RNA-seq datasets[13] (GSE148058; see below), and previously reported INS DRiPs[18]. PEAKS PTM search was performed with all 314 built-in modifications and with oxidation (M), acetylation (N-terminal), deamidation, and glutathionylation as variable modifications for proteasome inhibition experiments. Some specific PTM searches were run by setting the PTM of interest as a variable modification in both the *de-novo* and database search. The false discovery rate (FDR) was calculated with a decoy database search integrated into PEAKS and set to 1% at the peptide level for all samples. A 5% FDR was allowed when looking for mRNA variants reported in human islets. Label-free quantification of peptide abundance was performed using Progenesis QI v2 (Waters) for chromatographic alignment, normalization, and determination of ion abundances based on the area. Data analysis was performed with Python, Perseus and Excel. Sequence clustering was generated by GibbsCluster. All other figures and statistical analyses were done using GraphPad Prism (v9.0).

## RNAseq datasets and analysis

RNAseq datasets from 6 individual preparations of primary human islets exposed or not to IFN-α for 18 h[13] were retrieved from GEO: GSE148058. RNAseq datasets from 5 HLA Class II$^{lo}$ and HLA Class II$^{hi}$ human mTEC prepations[40] were retrieved from GEO:GSE201719. Gene expression was quantified using Salmon version 0.13.2[68] with parameters "--seqBias –gcBias –validateMappings". GENCODE v31 (GRCh38) was chosen as the reference genome and indexed with the default k-mer values. The estimated number of reads obtained from Salmon were used as input to perform differential expression with DESeq2 1.24.0[68]. For each gene included in DESeq2's model, a $\log_2$ FC was computed, and a Wald test statistics was assessed with unadjusted and adjusted $p$-values. Transcripts were considered differentially expressed when presenting an FC > 1.50 and an adjusted $p < 0.05$. Only transcripts presenting > 0.5 transcripts per million (TPM) in at least 20% of samples were selected for further analysis.

For the RNAseq pipeline, 30,947 mRNAs (TPM > 0.5) were filtered based on:

a. A median TPM > 5 in islets, either under basal or inflammatory conditions, a cut-off selected based on the median TPM of known islet antigens (islet expression filter; $n = 12,594$).

b. A median TPM < 0.1 in mTECs (either HLA Class II$^{lo}$ or HLA Class II$^{hi}$), or a median TPM fold-decrease > 100 vs. islet (mTEC expression filter).

c. A median TPM fold-increase > 10 in islets compared to 12 control tissues (adipose tissue, breast, colon, heart, kidney, liver, lung, lymph node, ovary, prostate, skeletal muscle, white blood cells), using the Illumina BodyMap 2.0 dataset (islet enrichment filter). Tissues of neuroendocrine origin (brain, testis, adrenal gland, and thyroid) were excluded from this filtering.

d. Subsequent analyses focused on mRNA isoforms[14]. The predicted translation products were aligned using the R package Biostrings v2.52.0, with BLOSUM100 as the score matrix, and alternative aa sequences were defined by comparing the predicted aa sequence of each mRNA isoform with that of the reference (canonical) mRNA, taking as reference the longest and/or most prevalent mRNA isoform in islets (alternative sequence generation filter).

## Immunopeptidomics bioinformatics analysis

An in-house analysis pipeline was designed (Python v3.7) and applied to the PEAKS PTM output file. After discarding long (> 14-aa) sequences, those remaining were sorted according to the expression of their source proteins/genes in the pancreas. Briefly, peptides whose source genes were found in the pancreas at the RNA level, or at the protein

level with a high/medium degree of evidence according to the Human Protein Atlas (V18) or the Human Protein Reference Database (release 9) were retained. Previously published single-cell RNAseq datasets from human pancreatic islets[69] were re-processed to generate a list of genes enriched in endocrine cells, which was used to further select MS-identified peptides. Together, these steps defined the "β-cell-enriched expression" filter. In parallel, the spectra found in the PEAKS PTM search (matched to the database) were excluded from the PEAKS *de-novo* search output, and the remaining spectra were searched for *cis*-spliced peptide sequences. The 10 reported peptides per spectrum were investigated for putative forward or reverse *cis*-spliced sequences (i.e., generated by the ligation of two fragments in the order in which they occur in the parent protein or in reverse order, respectively). Sequences were then fed into the MARS algorithm[21] trained with the genome-templated sequences. The MARS output and the putative *cis*-spliced sequences were compared, and sequences found in both were retained and filtered based on their β-cell-enriched expression as above. NetMHCpan4.1a[22] was installed locally and used to predict binding to the HLA alleles expressed by ECN90 β-cells (A*02:01/03:01, B*40:01/49:01, C*03:04/07:01, E*01:01; cutoff score < 2). For PTM peptides, predictions were performed on native sequences. NetMHCpan ranking is also embedded in the MARS algorithm. A single HLA prediction was attributed to a peptide if the score difference with the second-best allele was ≥ 3-fold; the two best HLA predictions were otherwise assigned.

## RT-qPCR
Total RNA was isolated from ECN90 β-cells using RNeasy mini kit (Qiagen), and from islets using Quick-RNA microprep kit (Zymo Research, #R1054), with RNA concentration and purity assessed by Nanodrop (ThermoFisher). cDNA was synthesized from 500 ng total RNA using a superscript VILO synthesis kit (ThermoFisher, #11754050) prior to gene expression analysis by RT-qPCR (Roche LightCycler or ThermoFisher QuantStudio 3). QuantiTect SYBR Green (Qiagen) or Power SYBR Green PCR Master Mix (Applied Biosystems) were used along with primers reported for HLA-I genes[15] or designed with Primer3 software (Supplementary Table 6). Melting curves were analyzed to evaluate amplicon specificity before quantification. Gene expression was normalized using *GAPDH*, *ACTB,* and *PPIA* as housekeeping genes. The ΔCt comparative method for relative quantification was used.

## Protein synthesis analysis by puromycin incorporation
IFN-α- or IFN-γ-treated ECN90 β-cells were pretreated or not with cycloheximide (50 μg/mL; Sigma) for 1 h before pulsing during the last 10 min with puromycin (10 μg/mL; ThermoFisher). After 3 washes with ice-cold PBS, cells were harvested, paraformaldehyde-fixed, and stained with AF488-coupled anti-puromycin Ab (RRID:AB_2736875; 1:200) and APC-coupled anti-HLA-A/B/C/E Ab (RRID:AB_314879; 1:200) in PBS/BSA with 1% saponin (ThermoFisher, #C10424) prior to acquisition on a BD LSRFortessa.

Neo-synthetized puromycin-labeled proINS was analyzed in ECN90 β-cells lysed in RIPA buffer with protease/phosphatase inhibitors. Proteins (600 μg) were incubated with proINS Ab (RRID:AB_10949314; 500 ng/mL) overnight at 4 °C, followed by addition of protein A/G agarose beads (ThermoFisher) at room temperature for 3 h. Immunoprecipitated complexes were washed 4 times and eluted in Laemmli buffer (Bio-Rad). Low-molecular-weight protein western blot was performed using a 12% Bis-Tris gel (ThermoFisher) under denaturing conditions with an SDS running buffer[70]. Protein transfer to a PVDF membrane was performed using the iBlot 2 system (ThermoFisher) at 20 V for 4 min. Following the transfer, the membrane was fixed in 0.2% glutaraldehyde (Sigma) for 15 min at room temperature. After three washes in PBS/0.1% Tween-20 (PBS-T), the membrane underwent epitope unmasking by an initial boiling step followed by microwave treatment in 10 mM citrate buffer (pH 6.0) containing 1 mM EDTA and 0.05% Tween-20 for 10 min at 600 W. Once cooled, the membrane was quenched with 200 mM glycine in PBS-T for 10 min to neutralize residual aldehydes. Blocking was then performed in Tris-buffered saline with 0.1% Tween-20 (TBS-T) and 5% non-fat dry milk for 1 h at room temperature. PVDF membranes were incubated with anti-puromycin Ab (1:1,000) overnight at 4 °C and secondary anti-mouse IgG Ab RRID:AB_330924 (1:2000) at room temperature for 2 h and revealed by ThermoFisher SuperSignal West Dura and iBright imaging system. After stripping, membranes were incubated with proINS Ab (1:1000) overnight at 4 °C and processed as above.

## Ab validation
HLA-I⁻ K562 cells (ATCC, #CCL-243) transduced with different HLA-I alleles[29,30] were single-stained with the following Abs: HLA-A clone ARC0588 (RRID: AB_2849011; 1:100) with secondary goat anti-rabbit IgG Ab RRID:AB_2536097 (1:100); HLA-B clone JOAN-1 (RRID:AB_1076708; 1:100) with secondary goat anti-mouse IgG Ab RRID:AB_2536161 (1:100); HLA-C Ab DT-9 (RRID:AB_2739715; 1:50); HLA-A/B/C/E Ab W6/32 (RRID:AB_314873; 1:20), followed by acquisition on a Beckman CytoFLEX flow cytometer.

## Flow cytometry
Surface HLA-I expression was analyzed on ECN90 β-cells single-stained for 30 min with Live/Dead Violet (ThermoFisher, #L34958; 1:1,000) and Abs to: HLA-A2 clone BB7.2 (RRID:AB_3068052; 1:100) with secondary goat anti-mouse IgG Ab RRID:AB_2921066 (1:200); HLA-B clone JOAN-1 (RRID:AB_1076708; 1:100), HLA-C clone DT-9 (RRID:AB_2650941; 1:100), HLA-A/B/C/E clone W6/32 (RRID:AB_314871; 1:100) with secondary goat anti-mouse IgG Ab RRID:AB_2921066 (1:200) before acquisition on an LSRFortessa with BD FACSDiVa v9.0 software and analysis with FlowJo v10.10.

Primary human islets from IIDP were dissociated with TrypLE (ThermoFisher) for 10 min at 37 °C prior to staining with Live/Dead Fixable Near-IR (ThermoFisher, # L10119; 1/2000) and Abs BV421-conjugated HLA-A2 clone BB7.2 (RRID:AB_2721522; 1:100) and FITC-conjugated HLA-A/B/C/E clone W6/32 (RRID:AB_314872; 1:100); HLA-B clone JOAN-1 (RRID:AB_1076708; 1:100), HLA-C clone DT-9 (RRID:AB_2650941; 1:100) or isotype control; followed by AF488-conjugated goat anti-mouse IgG (RRID:AB_2921066; 1:200) and by intracellular staining with AF647-conjugated anti-insulin (RRID:AB_2739331; 1:100) and PE-conjugated anti-glucagon (RRID:AB_2739382; 1:200) Abs. Samples were acquired on a Cytek Aurora, data was unmixed using SpectroFlo v2.2, and unmixed files were analyzed on FlowJo.

## Western blotting
Cell pellets were lysed in RIPA buffer with protease inhibitor cocktail (Sigma) and total protein was quantified with Pierce BCA protein assay kit (ThermoFisher). Protein samples were separated by Bolt 4–12% Bis-Tris polyacrylamide gels (Invitrogen) under denaturing conditions and probed with Abs ARC0588 (HLA-A, RRID:AB_2849011; 1:1000) and HC10 (HLA-B, RRID:AB_2728622; 1:1000) overnight at 4 °C, followed by horseradish peroxidase-conjugated secondary goat anti-rabbit IgG RRID:AB_2687483 and horse anti-mouse IgG RRID:AB_330924 (1:10,000), respectively. Alpha-tubulin Ab RRID:AB_1210457 (1:1000, 1 h at room temperature) was used with RRID:AB_330924 (1:10,000) for loading control. Bands were quantified with ImageJ.

## Tissue immunofluorescence
For multiplex tissue immunofluorescence, formalin-fixed paraffin-embedded pancreas tissue sections (Supplementary Table 2) were from EADB (https://pancreatlas.org/; West of Scotland Research Ethics Committee, 15/WS/0258) or nPOD (https://npod.org/; University of Florida Health Center Institutional Review Board, #201600029).

Sections were baked at 60 °C for 1 h, dewaxed in Histo-Clear, rehydrated in degrading ethanol concentrations (100%, 95%, 70%) and fixed in 10% neutral-buffered formalin. Heat-induced epitope retrieval (HIER; 10 mM citrate, pH 6) was performed for epitope unmasking by placing sections in a pressure cooker in a microwave oven at full power for 20 min. The sections were then blocked with 5% normal goat serum and incubated with primary Ab, followed by probing with an appropriate OPAL fluorophore-conjugated secondary Ab (Akoya Biosciences, Supplementary Table 7). This was followed by a further HIER to remove the primary Ab before staining with the next primary/secondary Ab combination. The same steps (from blocking to epitope retrieval) were repeated 5 times (for each of the 6 primary/secondary Ab combinations). Sections were counterstained with DAPI and mounted for multispectral fluorescent microscopy using the Vectra Polaris slide scanner (Akoya Biosciences). Quantification was performed using the Indica HALO image analysis platform. The DenseNet classification module was used to identify endocrine and exocrine regions on the whole slide scan (resolution 2 μm/pixel, minimum object size 750 μm²). The identified endocrine regions (islets) were manually sorted into ICIs and IDIs. The HighPlex module was used to segment and phenotype islet cells; β-cells were defined as $INS^+GCG^-$ and α-cells as $INS^-GCG^+$. Cell object data were exported in Excel and analyzed in GraphPad Prism 10.

**Antigen recall and cytotoxicity assays with TCR transductants**
HLA-B40-restricted peptides (Supplementary Table 4) were screened on TCRs obtained from nPOD donors listed in Supplementary Table 3. To generate fluorescent reporter TCR transductants[29,30], 5KC T-cell hybridomas carrying transgenes for an NFAT-driven ZsGreen-1 reporter and human CD8 were transduced with murine stem cell virus-based retroviral vectors produced in Phoenix-ECO cells (ATCC, #CRL-3214). Gene blocks encoding TCRs recognizing an HLA-B40-restricted $PPI_{44-52}$ epitope (TCR 173.D12) or an HLA-A2-restricted $PPI_{15-24}$ epitope (TCR 1E6) were inserted by Gibson assembly using NEBuilder HiFi DNA assembly mix (New England Biolabs, #M5520AA). TCRs were re-expressed as chimeric TCRα/β pairs (i.e., with human variable regions and murine constant regions) linked by a porcine teschovirus-1 2 A (P2A) peptide (synthesized by Twist Bioscience). Wild-type or *INS* KO ECN90 β-cells pulsed or not with TCR-cognate peptides (100 μM, Synpeptide), were exposed for 24 h to IFN-α, IFN-γ or no cytokine as above before adding TCR transductants (T:β-cell ratio 1:2) for 6 h.

To generate primary TCR transductants, human $CD8^+$ T cells were enriched by magnetic negative selection (RRID:AB_2728716) from peripheral blood mononuclear cells (PBMCs; collected under approval 2021-A01619-32 of Ouest IV/Nantes Ethics Committee). They were pre-activated 48 h with CD3/CD28 Dynabeads (RRID:AB_2916088) prior to transduction using lentiviral plasmids encoding TCRs recognizing an HLA-B40-restricted NY-ESO-$1_{125-133}$ epitope (TCR#100)[71] or an HLA-A2-restricted CMV pp65$_{495-503}$ epitope (TCR#57). TCRs were re-expressed as chimeric human TCRα/β pairs linked by a P2A peptide on a murine TRAC/TRBC backbone (Twist Bioscience). Transduced T cells were magnetically enriched with biotin-conjugated murine TCRβ Ab (RRID: AB_394680) and expanded 6 more days with Proleukin.

For cytotoxicity assays[55], ECN90 β-cells were cultured with or without IFN-α for 24 h, stained with 1 μM CFSE or CellTrace Violet (CTV; ThermoFisher), and pulsed, respectively, for 1 h with 5 μM TCR-cognate peptide or peptide diluent. After washing, CFSE- and CTV-labeled β-cells were mixed in equal numbers (total $10^5$ cells/well) and cultured for 18 h in sextuplicate in 96-well flat-bottom plates, alone or with T cells ($4 \times 10^4$/well; effector-to-target ratio 1:2.5). After washing, cells were stained with Live/Dead Fixable Far Red (ThermoFisher, #L34973) and acquired on a BD LSRFortessa cytometer (Supplementary Fig. 9a). β-cells were analyzed after gating on Live/Dead$^-$ events and separation of CFSE$^+$ (peptide-pulsed) and CTV$^+$ (unpulsed) populations. Percent peptide-specific lysis is expressed as the ratio of live

CFSE$^+$/CTV$^+$ cells normalized to the same ratio in wells containing β-cell targets alone.

**Antigen recall on islet-infiltrating CD8$^+$ T cells**
Isolated islets or live pancreas slices (150-μm thick) were provided by nPOD (donors listed in Supplementary Table 3). Islets were isolated by tissue digestion with 1 mg/mL collagenase-P (Sigma, #11213857001) in PBS at 37 °C under shaking with a magnetic stir bar and visual monitoring for tissue dispersion, followed by washing and hand-picking. To generate islet-derived T-cell lines[31], islet-infiltrating T cells were recovered and expanded from isolated islets (nPOD donors 6342 and 6480) or live pancreas slices (nPOD donors 6536, 6563, 6566) (Supplementary Table 3). For the latter, 6-10 slices per donor ($\sim 1 \times 1\,mm \times 150-200\,\mu m$) were digested with collagenase-P (1 mg/mL) for 8–10 min at 37 °C under shaking. After washing, individual islets were hand-picked from the pancreas slice digest (or from isolated islets for donors 6342, 6480) and distributed in round-bottom 96-well plates (Corning, #3799; 1 islet/well) with $1.5 \times 10^5$ irradiated allogeneic PBMCs as feeders in AIM-V medium (ThermoFisher, #12055083) supplemented with 5% heat-inactivated human male AB serum (Grifols Bio Supplies), 2 mM L-glutamine, 5 mM HEPES, 100 μ/mL penicillin, 100 mg/mL streptomycin, 0.1 mM non-essential aa, 1 mM sodium pyruvate (all from ThermoFisher). For initial T-cell stimulation and expansion, medium contained soluble anti-CD3 (RRID:AB_395736; 2.5 μg/mL) and anti-CD28 (RRID:AB_396068; 2.5 μg/mL), a blocking anti-Fas Ab (RRID:AB_10596808; 1 μg/mL), anti-PD-1 Ab (RRID:AB_10897007; 1 μg/mL), mifepristone (100 nM; Invitrogen, #H11001), Proleukin (20 μ/mL), and IL-15 (10 ng/mL; ThermoFisher, #200-15-UG). After 16-20 days of culture with splitting as needed, medium change (without Abs), and fresh cytokines every 2-3 days, T-cell lines were collected, phenotyped by flow cytometry for T-cell subsets, and cryopreserved at passage 1–3. Lines selected herein contained > 20% CD8$^+$ T cells.

Irradiated (5000 rads) HLA-B40-transduced K562 cells (from M. Nakayama) were plated at 3000 cells/well and pulsed for 2 h at 30 μg/mL per peptide with HLA-B40-restricted peptide pools (Supplementary Table 4) or a negative control pool of peptides binding to irrelevant HLA-I alleles (Supplementary Table 5)[19,72]. After washing, they were then co-cultured for 48–72 h with individual islet-derived T-cell lines (or pooled lines when needed to reach suitable T-cell numbers) at 75,000 cells/well in triplicate wells of round-bottom 96-well plates. T cells alone were cultured with and without plate-bound anti-CD3/CD28 as a positive control for functional, viable T cells. Supernatants were collected, and IFN-γ was measured by a cytometric bead array (BD, #561515; 274 fg/ml lower detection limit). The reactivity of selected T-cell lines was deconvoluted for individual peptides as above.

**Statistical analysis**
Statistical details of experiments can be found in the legends of each figure. A two-tailed $p < 0.05$ was used to define statistical significance.

**Reporting summary**
Further information on research design is available in the Nature Portfolio Reporting Summary linked to this article.

## Data availability
The mass spectrometry data generated in this study have been deposited in the ProteomeXchange Consortium via the PRIDE partner repository under PXD045265 (ECN90 β-cells) [https://proteomecentral.proteomexchange.org/cgi/GetDataset?ID = PXD045265] and PXD045211 (primary human islets) [https://proteomecentral.proteomexchange.org/cgi/GetDataset?ID = PXD045211]. The public RNAseq datasets utilized in this study are available in the GEO repository under GSE148058 (human islets) and

GSE201719 (human mTECs). All data are included in the Supplementary Information or available from the authors, as are unique reagents used in this Article. The raw numbers for charts and graphs are available in the Source Data file whenever possible. Raw flow cytometry data and tissue immunofluorescence images can be obtained without restriction from R. Mallone and S.J. Richardson, respectively. Source data are provided in this paper.

## Code availability

The code generated for immunopeptidomics bioinformatics analysis has been deposited in the Zenodo repository under DOI:10.5281/zenodo.14496237 [https://zenodo.org/records/14496237] and is available under a GNU GPLv3 open-source license.

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

## Acknowledgements

We gratefully acknowledge L. Bailly (myBrain Technologies, Paris) for his help in developing the bioinformatics analysis pipeline; O. Mayeur, S. Salignac, A. El Kiyane (Cochin Institute, Paris) and J. Cobb (IBRI, Indianapolis) for technical assistance; M. Peakman (King's College, London) for providing the 1E6 TCR sequence and N. Hirano (University Health Network, Toronto) for providing the B40/NY-ESO-1 TCR sequence; and the Cochin Institute CYBIO Flow Cytometry Core Facility for assistance with cell analysis. This work was funded by The Leona M. and Harry B. Helmsley Charitable Trust project 1901-03689, Agence Nationale de la Recherche project ANR-19-CE15-0014-01, Fondation pour la Recherche Medicale project EQU20193007831 (R.M.); European Foundation for the Study of Diabetes EFSD/JDRF/Lilly European Program in Type 1 Diabetes Research 2019 (N.T.); JDRF Postdoctoral Fellowship 3-PDF-2020-942-A-N (Z.Z.); EFSD/Lilly Young Investigator Research Award Program 2023 (F.S.); NIH grants U01 DK104218-04 and UC4 DK116284 (S.C.K.); NIH grants R01 DK099317, P30 DK116073 (M.N.); Steve Morgan Foundation Grand Challenge Senior Research Fellowship 22/0006504 (S.J.R.); JDRF grants 3-SRA-2022-1201-S-B (1 and 2), Welbio-FNRS (Fonds National de la Recherche Scientifique) project WELBIO-CR-2019C-04, the NIH Human Islet Research Network Consortium on Beta Cell Death & Survival (HIRN-CBDS) project U01 DK127786, NIH/NIDDK grants RO1DK126444 and RO1DK133881-01 (D.L.E.); the Innovative Medicines Initiative 2 Joint Undertaking under grant agreements 115797 and 945268 (INNODIA and INNODIA HARVEST), which receive support from the EU Horizon 2020 program, JDRF, and The Leona M. & Harry B. Helmsley Charitable Trust

(R.M., D.L.E., R.S., and S.J.R.). Human islets were provided by the NIDDK-funded IIDP (RRID:SCR_014387) at the City of Hope (NIH grant 2UC4DK098085). This research was performed with the support of the Network for Pancreatic Organ Donors with Diabetes (nPOD; RRID:SCR_014641), a collaborative T1D research project supported by Breakthrough T1D and The Leona M. and Harry B. Helmsley Charitable Trust (grant #3-SRA-2023-1417-S-B). The content and views expressed are the responsibility of the authors and do not necessarily reflect the official view of nPOD. Organ Procurement Organizations (OPO) part-nering with nPOD to provide research resources are listed at https://npod.org/for-partners/npod-partners/.

## Author contributions

A.C. performed immunopeptidomics experiments, analyzed data, par-ticipated in the conceptualization and supervision of the study, wrote the first draft, and edited subsequent drafts. F.S. and Z.Z. performed HLA-I expression and T-cell transductant experiments, analyzed data, and participated in the conceptualization of the study. J.P.-H., O.B.M., J.A., and D.P. performed HLA-I expression experiments and analyzed data. C.L. and S.J.R. conceived and performed pancreas tissue immuno-fluorescence experiments and analyzed data. A.M. and S.C.K. generated T-cell lines, performed T-cell experiments, and analyzed data. M.O. characterized β-cell changes upon IFN treatment and analyzed data. H.L. performed the analysis of *cis*-spliced peptides. R.P. and A.N. parti-cipated in immunopeptidomics experiments and data analysis. B.B. generated and validated T-cell transductants and performed RT-qPCR experiments. M.L.C. and D.L.E. generated the dataset of mRNA splice variants. D.L.E., M.G., and S.R.H. participated in the conceptualization of the study. A.A., L.L., and M.N. participated in the generation of T-cell transductants and validation of HLA-I Abs. F.K. and R.S. contributed wild-type and *INS* KO ECN90 β-cells and edited the manuscript. L.M. and P.M. contributed primary human islets. S.Y. contributed to study supervision, funding acquisition, and manuscript editing. N.T. supervised the study, provided resources, and acquired funding. R.M. conceived and super-vised the study, provided resources, acquired funding, and wrote the final version of the manuscript.

## Competing interests

The authors declare no competing interests.

## Additional information

[1]Université Paris Cité, Institut Cochin, CNRS, INSERM, Paris, France. [2]Department of Nutrition and Health, Valencian International University (VIU), Valencia, Spain. [3]Islet Biology Group, Exeter Centre of Excellence in Diabetes Research, University of Exeter Medical School, Exeter, UK. [4]Diabetes Center of Excellence, Department of Medicine, University of Massachusetts Chan Medical School, Worcester, MA, USA. [5]Centre for Immuno-Oncology, Nuffield Department of Medicine, University of Oxford, Oxford, UK. [6]ULB Center for Diabetes Research, Université Libre de Bruxelles, Brussels, Belgium. [7]Indiana Biosciences Research Institute, Indianapolis, IN, USA. [8]Department of Health Technology, Technical University of Denmark, Copenhagen, Denmark. [9]Barbara Davis Center for Diabetes, University of Colorado School of Medicine, Aurora, CO, USA. [10]Department of Clinical and Experimental Medicine, University of Pisa, Pisa, Italy. [11]Assistance Publique Hôpitaux de Paris, Service de Diabétologie et Immunologie Clinique, Cochin Hospital, Paris, France. [12]These authors contributed equally: Fatoumata Samassa, Zhicheng Zhou. ✉e-mail: roberto.mallone@inserm.fr

