## [Transparent Peer Review file · Nature Communications]

Interferon- α promotes HLA-B-restricted presentation of conventional and alternative antigens in human pancreatic β -cells

Corresponding Author: Professor Roberto Mallone

Version 0:

Reviewer comments:

Reviewer #1

(Remarks to the Author)
Review NCOMMS-23-40219

The manuscript titled "Interferon- α promotes neo-antigen formation and preferential HLA-B-restricted antigen presentation in pancreatic β -cells" by Carré and Malone examines the repertoire of HLA-class I peptides presented by beta cells prior and after IFN α stimulation. The work is thorough and well done and expanded to the description of CD8 T cell responses from patients. Some modifications, major and minor must be implemented before publication.

Major issues

In the introduction, there are a couple of misstatements that will confuse some readers. The first one is to present IFN α as the sole IFN important for progression. Apart for the essential role of IFN γ (that should be mentioned), IFN β is also important. It is only in transgenic INS-antigen viral models that α has been shown to be essential. In the NOD, both α and β are important and β probably more so. In human, the situation is not settled. Second issue, it should not be said that "type 1 IFNs upregulate expression of human leukocyte antigen (HLA) Class I (HLA-I) and Class II along with costimulatory molecules³". Type 1 IFNs regulate only class I as there is no type 1 IFN response element of the class II genes, only γ . The HC-10 antibody that is used in Figure 5 to trace HLA-B expression is not HLA-B specific but reactive to the heavy chain of all three class I, A, B, and C. An anti-HLA-B polyclonal antibody must be used to reach any conclusion. I do understand why the work is limited to a single IFN, but it should not obscure the complexity of the in vivo milieu to which beta cells will be exposed. A nice complementary experiment would simply be to compare HLA-A and B expression (by FACS and qPCR) upon single stimulation with α , β , and γ , and their combinations; one could also add TNF and IL1 β as both have been shown in the islet. This experiment should come in supplementation of figure 5.

Minor issues

The fact that the entire study is performed on a beta cell line should be mentioned earlier than in the Results section. It should also be validated by a confirmation of the overexpression of HLA-B over HLA-A in primary human islets (by flow cytometry) as this conclusion is one of the most important ones of the paper if proven.

The statement about PTM and spliced peptides should be softened and put in context, especially given the large number of dropouts after comparison with the synthetic peptides. The field is very controversial because most "spliced peptides" or PTM peptides have been diagnosed on mass only. Here the work is well done but the attrition is high. Given the minor representation of those modified peptides in the overall peptidome, their biology needs to be taken with a grain of salt.

The discussion is too long and unfocused.

Reviewer #2

(Remarks to the Author)

The article by Carré A et al., shows that IFN- α , the earliest cytokine signature in individuals at risk for T1D, profoundly impacts the repertoire of HLA-I-bound peptides presented by pancreatic β -cells. Using immunopeptidomics, this study reveals that IFN- α enhances HLA-I expression and peptide presentation, generating neo-sequences through mRNA splicing, post-translational modifications, and protein cis-splicing. HLA-B is favored over HLA-A, leading to an increased activation of HLA-B-restricted CD8+ T cells. This HLA-B skewing is observed in T1D islets, suggesting its relevance to T1D pathogenesis.

The results include the significant impact of IFN- α on HLA-I-bound peptide presentation in pancreatic β -cells, particularly the preference for HLA-B over HLA-A and the potential relevance to T1D pathogenesis. The work is of significant importance to the field and expands our understanding of T1D immunogenicity. It aligns with established literature on IFN- α 's role but offers novel insights into HLA-B presentation. Interpretation is limited by the small sample size; however, reported data adequately support the conclusions. The methodology is sound and meets expected standards in the field.

Major criticisms:

- a) The role of HLA-B peptides in T1D immunogenicity should be addressed more comprehensively. The authors should explore whether HLA-B peptides induced by IFN- α are deleterious or protective in islet immunogenicity. T-cell reactivity studies (Figure 6) should be complemented with *in vitro* experiments to determine if HLA-B40-restricted PPI TCRs lead to increased islet killing. The discussion on the potential impact of HLA-B peptide presentation should be expanded.
- b) The relationship between HLA-A and HLA-B responses to IFN- α needs clarification. Does IFN- α induce an additional or rather an alternative pathway of immunogenicity mediated by HLA-B?
- c) The presentation of data in Figure 1 could be clarified. Specify whether the bars in Figure 1A represent the median frequency of peptide length obtained from all four replicates or if they are representative images of one experiment. In the Venn diagram of fig 1B, the "basal" peptides are the one exclusively found in all four replicates of basal condition? How do you categorize peptides found as basal in one experiment and found in both basal and IFN- α treatment in another experiment within the Venn diagram?

Reviewer #3

(Remarks to the Author)

Carré et al. report here a proteogenomic study of pancreatic beta cells exposed to IFN-alpha. The rationale of the study is that IFN-a is the earliest cytokine observed in individuals at risk of type 1 diabetes, an autoimmune reaction notably mediated by T-cell attacks against beta cells. They study the immunopeptidome of the ECN90 cell line exposed or not to IFN-a and discover a general increase of the number of MHC epitopes. This increase is subsequently ascribed to a preferential upregulation of HLA-B molecules. This increased expression induces the generation of neopeptides and immunogenic peptides that are recognized by CD8 T cells in patients. Overall, the article reports findings that are of undeniable significance to the field. The study is also well conducted and methodologically robust. However, I have some reserve regarding the novelty, the clarity and the mechanistic insights provided by the article. My specific comments are listed below:

- The novelty of the article appears to be relatively limited, and the significance of its findings is either poorly exposed or unclear. In 2018, the authors published a nearly identical study that reported the upregulation of HLA molecules by IFN γ in the ECN90 cell line, which is the same cell line used in this study. Furthermore, the previous study also concluded that IFN γ upregulates MHC-I molecules, leading to the generation of neoantigens that are targeted by CD8 T cells (PMID 30078552). The main distinction between the current article and the previous one is that the authors investigated the effects of IFN β instead of IFN γ , and they conducted HLA-A/B/C-specific analyses that were absent in the earlier work. It is essential for the authors to clearly discuss the differences between the present study and the previous one. They could perform dedicated analyses comparing the current results to those of the earlier study and delve into the molecular disparities between the two cytokines, IFN γ and IFN β . For example, the authors could assess whether some of their current findings are specific to IFN β . Questions that could be addressed in this context include: Do similar preferential effects on HLA-B molecules and epitopes occur after treatment with IFN γ ? Is the impact of both cytokines on the immunopeptidome comparable? Do these effects correlate with changes in the transcriptome, or are they the result of post-translational alterations? Are there signaling differences in both molecules that could explain the observed distinctions? Is there any relevance to the pathophysiology of Type 1 Diabetes (T1D)?

While the authors have briefly touched upon some of these considerations in the discussion section, providing data to support these points seems feasible and could significantly enhance the novelty of the study.

- The term "neo-antigen" used in the title is not defined in the text. Sometimes the terms "neo-epitopes" and "neo-sequences" are also used, are they synonyms? I assume that the definition of neo-antigen is: "any peptide presented by IFN-a treated cells, and not presented by control cells". In this case, the authors do not provide full details about the neo-antigens (which are yet one of the main topics of the article). They only mention that some (18) PTM or spliced peptides are neo-antigens. However, they also mention (line 132) that 35.6% of peptides are specifically presented by IFN-a treated cells. What is the composition of these neo-peptides? Do they derive from proteins that are supposedly immunogenic? Why are the PTM/spliced peptides so important to deserve a detailed/focused analysis while the vast majority of peptides is "ignored"? Are they supposed to be more immunogenic? Overall, it is not clear to me why these neo-antigens are special/interesting.

- Regarding neo-antigens deriving from PTM or splicing, the authors should consider that many genomic loci other than exons can generate MHC epitopes (PMID 26728094). Therefore, it is possible that the genomic origin of the neo-antigens that they identify is not the one they assume but could be a genomic sequence absent from their MS database or transcriptome mapping tools. To illustrate this, a recent study (PMID 37582761) has shown that many MHC epitopes assumed to derive from proteasomal cis-splicing (PMID 27846572, this study is not cited to my great surprise, while it is the founding study about peptide cis-splicing), can in fact derive from genomic-templated (non-spliced) loci. A possible verification analysis here would be to check the absence of a genomic locus capable of coding for their neo-antigens (or their leucine/isoleucine variants) with the website <https://bamquery.irc.ca/search> (the tool used in PMID 37582761)

- The rationale for the proteasome-related immunopeptidomic analyses is not clear to me, and the interest of the conclusion (lines 225-226) is not clear either. Is it not perfectly expected that the immunopeptidome is shaped by the activity of the constitutive- and immuno-proteasomes? This part seems unrelated to the rest of the manuscript. To reconnect these observations to the rest, the authors could check whether IFN α -induced peptide (neoantigens) derive more frequently from chymotryptic cleavage since IFN- α upregulates the immunoproteasome.

- My previous comment illustrates well a more general comment: many analyses seem to be only for descriptive purpose, and to be performed without a clear rationale/conclusion to be taken. This results in a manuscript that is hard to read and a general message that is diluted in observations that seem irrelevant to the main message. I would encourage the authors to make their manuscript more focused/straightforward.

- It is not clear how IFN α increases the expression of HLA-B without affecting dramatically the other alleles (HLA-A/C). The authors should offer a mechanism, supported by experimental data, to explain this.

- Do the authors have an explanation to the immunogenicity of ABCC8 and CHGA-derived peptides? These are normal proteins (not mutated or dysfunctional) that are part of the normal biological processes of the organism. Their recognition by CD8 T cells is not trivial.

- This latter comment raises more general questions that could help the authors clarify/develop their manuscript:

A) Do the authors have an explanation for the origin of the immune reaction targeting HLA-B-presented peptides?

B) Is it expected that HLA-B overexpression by IFN- α automatically results in the generation of peptides recognized by CD8 T cells, or is this phenomenon restricted to individuals predisposed to T1D?

C) In case they assume that HLA-B overexpression leads to greater recognition of HLA-B-presented peptides by CD8 T cells in healthy individuals (response to previous question), do they believe this also applies to other HLA alleles?

D) If the response is yes in the previous question, how do they explain that normal peptides become immunogenic simply through the upregulation of the HLA molecule presenting them?

E) Finally, how is it possible that any anti-viral immune reaction (producing IFN α) in the pancreas does not induce autoimmune reactions against beta cells if IFN α induces HLA-B molecules, triggering the activation of CD8 T cells attacking beta cells?

- Line 112, the authors indicate that their reference database has been complemented with translated mRNA isoforms enriched in human islets exposed or not to IFN α . It is not clear to me what are these isoforms and why they were absent from the reference database. Likewise, why are the open-reading frames of insulin not included in this reference database?

- It is not clear how many mTECs were sequenced and used in the selection pipeline of neoantigens in line 608-621 (if this is indeed the selection filters for neoantigens, this is not clear either).

- I could not find the repository for mTEC RNA-seq data (fastq files), the authors should indicate where reviewers and readers can find these data to ensure the reproducibility of their pipeline.

- The authors indicate on line 595 that "RNAs from 6 individual preparations of primary human islets exposed or not to IFN- α for 18h [...] were sequenced on an Illumina HiSeq 2000 at high depth" and on line 606 that "Datasets have been deposited under GEO:GSE148058.". This suggests that the RNA sequencing and data deposition were performed for the need of the present study, by its main authors. However, the GSE148058 dataset was deposited by the lab of Maikel Luis Colli (co-author of this study) for the need of their study in 2020 (PMID 32444635). This is misleading then and should be rephrased to clearly acknowledge that the generation of the data was made in another study.

Version 1:

Reviewer comments:

Reviewer #1

(Remarks to the Author)

A substantial amount of work has been put into this revision and I am grateful to the other for this effort. While the criticism about HC10, the main anti-HLA-B antibody used for validation in the study, has been partially answered, its exclusive use in tissue section is still short of alleviating all concerns. I had suggested the use of anti-HLA-B polyclonal antibodies for confirmation. I am saddened that this step has not been taken.

(Remarks on code availability)

Reviewer #4

(Remarks to the Author)

(Remarks on code availability)

Reviewer #5

(Remarks to the Author)

Figure 1 has been clarified, as requested by Reviewer #2, and the addition of Discussion and Supplementary Fig.11 nicely highlight the potential pathomechanistic significance of the findings in the manuscript.

As for the request of adding additional evidence for the pathogenic role of islet peptides presented by HLA-B and a demonstration of killing potential of HLA-B40 restricted T cells, the authors have 1) performed additional analysis of islet-derived T-cell lines, demonstrating the recognition of multiple islet epitopes in the context of HLA-B40 and 2) indirectly demonstrated that IFN- α -induced upregulation of HLA-B expression leads to increased cytolytic activity using non-islet-specific TCRs and peptide-pulsed ECN90 cells. These expand on the original experiments, but naturally do not still conclusively demonstrate a pathogenic role of HLA-B epitopes in T1D pathogenesis. This would understandably require complex experiments that are beyond the scope of the current manuscript.

I would still suggest to add this caveat shortly in the Discussion, i.e., that the current study does not conclusively demonstrate that the increased presentation of HLA-B-restricted islet peptides leads to increased killing of beta cells and consequently further research would be needed to clarify whether the increased presentation of islet peptides by HLA-B has a potential pathogenic or protective effect on T1D pathogenesis.

(Remarks on code availability)

REPLIES TO REVIEWER COMMENTS

Reviewer #1 (expert in antigen presentation and MHC molecules)

Reviewer: The manuscript titled "Interferon- α promotes neo-antigen formation and preferential HLA-B-restricted antigen presentation in pancreatic β -cells" by Carré and Malone examines the repertoire of HLA-class I peptides presented by beta cells prior and after IFN α stimulation. The work is thorough and well done and expanded to the description of CD8 T cell responses from patients. Some modifications, major and minor must be implemented before publication.

Reply: We thank the Reviewer for his/her positive evaluation.

Reviewer: In the introduction, there are a couple of misstatements that will confuse some readers. The first one is to present IFN α as the sole IFN important for progression. Apart for the essential role of IFN γ (that should be mentioned), IFN β is also important. It is only in transgenic INS-antigen viral models that α has been shown to be essential. In the NOD, both α and β are important and β probably more so. In human, the situation is not settled. Second issue, it should not be said that "type 1 IFNs upregulate expression of human leukocyte antigen (HLA) Class I (HLA-I) and Class II along with costimulatory molecules³". Type 1 IFNs regulate only class I as there is no type 1 IFN response element of the class II genes, only γ .

Reply: We thank the Reviewer for pointing out these mis-statements in the Introduction. They have now been corrected.

Reviewer: The HC-10 antibody that is used in Figure 5 to trace HLA-B expression is not HLA-B specific but reactive to the heavy chain of all three class I, A, B, and C. An anti-HLA-B polyclonal antibody must be used to reach any conclusion.

Reply: Although not fully HLA-B-specific, the reactivity of the HC10 antibody has been extensively characterized at the precise epitope level (Perosa et al, J Immunol 2003, now quoted in the manuscript). This epitope mapping also gives reason for why this antibody stains β 2m-free (or loosely attached) HLA Class I molecules, and likely explains why it works in Western blot and immunofluorescence, but not in flow cytometry. In summary, apart from a minor cross-reactivity with HLA-C7 (which we also report in the manuscript), the relevant HLA-A cross-reactivities are: A*25:01, A*26:01, A*30:01, A31, A*33:03, A*34:01, A66, A*68:02, A*69:01, A*80:01 and, possibly, A10 and A28.

Looking at the HLA-A haplotype of the nPOD donors used for pancreas tissue staining (now listed in Supplementary Table 2), these cross-reactivities may impact conclusions for T1D case 6380 (A*33:03, A*68:02) and non-diabetic cases 6278 (A*68:02) and 6462 (A*25:01). To exclude this possibility, we extended our studies to 3 additional cases (1 T1D and 2 non-diabetic) not harboring these cross-reactive HLA-A alleles, and repeated the analysis by including all cases, or by excluding the three cases mentioned above (represented with crossed symbols in new Fig. 5d). Both analyses conclude that HLA-B is preferentially hyper-expressed, thus dismissing this possibility.

Reviewer: I do understand why the work is limited to a single IFN, but it should not obscure the complexity of the in vivo milieu to which beta cells will be exposed. A nice complementary experiment would simply be to compare HLA-A and B expression (by FACS and qPCR) upon single stimulation with α , β , and γ , and their combinations; one could also add TNF and IL1 β as both have been shown in the islet. This experiment should come in supplementation of figure 5.

Reply: We have performed these experiments using different cytokines and combinations thereof: IFN- α , IFN- β , IFN- γ , IFN- λ , IFN- α /IFN- β , IFN- α /IFN- γ , IFN- β /IFN- γ . These results are presented in new Fig. 4a-b, showing that ECN90 β cells upregulate HLA Class I in response to all IFNs, barring a marginal effect of

IFN- λ , TNF- α and/or IL-1 β (alone or in combination with IFNs) had no effect. More importantly, this HLA Class I upregulation invariably favors HLA-B and, to a lesser extent, HLA-C, over HLA-A.

Reviewer: The fact that the entire study is performed on a beta cell line should be mentioned earlier than in the Results section. It should also be validated by a confirmation of the overexpression of HLA-B over HLA-A in primary human islets (by flow cytometry) as this conclusion is one of the most important ones of the paper if proven.

Reply: We have done these validation experiments on primary islets from 7 non-diabetic organ donors by flow cytometry (new Fig. 4e), with 3 preparations also analyzed by qPCR (new Fig. 4d). Although with some expected larger variability compared to the cell line model, these experiments confirm our observations on ECN90 β cells.

Reviewer: The statement about PTM and spliced peptides should be softened and put in context, especially given the large number of dropouts after comparison with the synthetic peptides. The field is very controversial because most “spliced peptides” or PTM peptides have been diagnosed on mass only. Here the work is well done but the attrition is high. Given the minor representation of those modified peptides in the overall peptidome, their biology needs to be taken with a grain of salt.

Reply: We agree with the Reviewer that the attrition is high. However, the stringent criteria used to define bona fide biological PTMs or spliced peptides should be taken into account, meaning that those peptides filtered out may well exist, but cannot be confidently assigned. We have introduced a comment in the Discussion: “Although unavoidable experimental artifacts preclude a comprehensive assessment of the PTM peptidome, most of the stringently validated PTM peptides (11/12), *cis*-spliced peptides (6/9) and the INS-205 mRNA-spliced peptide were found exclusively after IFN- α treatment”.

Reviewer: The discussion is too long and unfocused.

Reply: We have shortened and refocused the Discussion accordingly, and expanded some sections as requested by other Reviewers.

Reviewer #2 (expert in the immunology of type-1 diabetes)

Reviewer: The article by Carré A et al., shows that IFN- α , the earliest cytokine signature in individuals at risk for T1D, profoundly impacts the repertoire of HLA-I-bound peptides presented by pancreatic β -cells. Using immunopeptidomics, this study reveals that IFN- α enhances HLA-I expression and peptide presentation, generating neo-sequences through mRNA splicing, post-translational modifications, and protein *cis*-splicing. HLA-B is favored over HLA-A, leading to an increased activation of HLA-B-restricted CD8+ T cells. This HLA-B skewing is observed in T1D islets, suggesting its relevance to T1D pathogenesis. The results include the significant impact of IFN- α on HLA-I-bound peptide presentation in pancreatic β -cells, particularly the preference for HLA-B over HLA-A and the potential relevance to T1D pathogenesis. The work is of significant importance to the field and expands our understanding of T1D immunogenicity. It aligns with established literature on IFN- α 's role but offers novel insights into HLA-B presentation. Interpretation is limited the small sample size; however, reported data adequately support the conclusions. The methodology is sound and meets expected standards in the field.

Reply: We thank the Reviewer for his/her positive evaluation.

Reviewer: a) The role of HLA-B peptides in T1D immunogenicity should be addressed more comprehensively. The authors should explore whether HLA-B peptides induced by IFN- α are deleterious or protective in islet immunogenicity. T-cell reactivity studies (Figure 6) should be complemented with in

vitro experiments to determine if HLA-B40-restricted PPI TCRs lead to increased islet killing. The discussion on the potential impact of HLA-B peptide presentation should be expanded.

Reply: We thank the Reviewer for raising this important point. First, we expanded in-vitro studies to deconvolute the antigen specificity of islet-infiltrating CD8⁺ T-cell lines testing positive to peptide pools using individual peptides. These experiments are presented in new Fig. 6e-f-g and identified several additional hits at the individual epitope level. They thus provide evidence that HLA-B40-restricted CD8⁺ T cells represent a significant fraction of immune infiltrates in the islets.

Second, the two HLA-A2-restricted and HLA-B40-restricted PPI epitopes and cognate TCRs shown in the previous figure (T-cell activation readout) were not suitable for the proposed β -cell killing experiments, due to the different degree of presentation of these particular peptides (Fig. 6c). We have therefore performed these experiments using TCRs reactive to non- β -cell epitopes on ECN90 β -cell pulsed with peptides, in order to normalize peptide presentation and isolate the contribution of HLA-B upregulation. These results are shown in new Fig. 6d, which documents an increased killing of β -cells exposed to IFN- α after challenging with HLA-B40-restricted but not HLA-A2-restricted CD8⁺ T cells. The discussion on the potential impact of HLA-B peptide presentation has also been expanded, and a new Supplementary Fig. 11 has been added to depict the model.

Reviewer: b) The relationship between HLA-A and HLA-B responses to IFN- α needs clarification. Does IFN- α induce an additional or rather an alternative pathway of immunogenicity mediated by HLA-B?

Reply: Our results do not dismiss the importance of HLA-A-restricted peptides. They rather highlight the additional contribution of HLA-B-restricted ones, which, given its reliance on IFN- α and more generally IFNs, may be critical to unmask β -cells for immune recognition and flip the balance toward loss of immune tolerance. In other words, preferential HLA-B upregulation may represent a previously unappreciated mechanism of loss of immune ignorance and epitope spreading. We have now clarified this point in the Discussion (page 18 and Supplementary Fig. 11).

Reviewer: c) The presentation of data in Figure 1 could be clarified. Specify whether the bars in Figure 1A represent the median frequency of peptide length obtained from all four replicates or if they are representative images of one experiment. In the Venn diagram of fig 1B, the “basal” peptides are the one exclusively found in all four replicates of basal condition? How do you categorize peptides found as basal in one experiment and found in both basal and IFN- α treatment in another experiment within the Venn diagram?

Reply: The bars in Fig. 1a represent the cumulative peptide counts obtained from all 4 replicates – this is now clarified in the figure legend. The “basal” peptides in the Venn diagram of Fig. 1b are the ones identified in at least one replicate of the basal condition. A peptide found in basal in one experiment and in both basal and IFN- α conditions in another experiment would be considered as “both” in the Venn diagram. This is now also specified in the figure legend. While we opted for not being too stringent when categorizing peptides, we kept track of the details about conditions and number of replicates in which peptides were identified in Supplementary Data 1, 2 and 3 (columns C-D).

Reviewer #3 (expert in immunopeptidomics)

Reviewer: Carré et al. report here a proteogenomic study of pancreatic beta cells exposed to IFN-alpha. The rationale of the study is that IFN-a is the earliest cytokine observed in individuals at risk of type 1 diabetes, an autoimmune reaction notably mediated by T-cell attacks against beta cells. They study the immunopeptidome of the ECN90 cell line exposed or not to IFN-a and discover a general increase of the number of MHC epitopes. This increase is subsequently ascribed to a preferential upregulation of HLA-B

molecules. This increased expression induces the generation of neopeptides and immunogenic peptides that are recognized by CD8 T cells in patients. Overall, the article reports findings that are of undeniable significance to the field. The study is also well conducted and methodologically robust. However, I have some reserve regarding the novelty, the clarity and the mechanistic insights provided by the article. My specific comments are listed below.

Reply: We thank the reviewer for his/her positive feedback and for pointing out some limitations, which we address below.

Reviewer: - The novelty of the article appears to be relatively limited, and the significance of its findings is either poorly exposed or unclear. In 2018, the authors published a nearly identical study that reported the upregulation of HLA molecules by IFN γ in the ECN90 cell line, which is the same cell line used in this study. Furthermore, the previous study also concluded that IFN γ upregulates MHC-I molecules, leading to the generation of neoantigens that are targeted by CD8 T cells (PMID 30078552). The main distinction between the current article and the previous one is that the authors investigated the effects of IFN β instead of IFN γ , and they conducted HLA-A/B/C-specific analyses that were absent in the earlier work. It is essential for the authors to clearly discuss the differences between the present study and the previous one. They could perform dedicated analyses comparing the current results to those of the earlier study and delve into the molecular disparities between the two cytokines, IFN γ and IFN β . For example, the authors could assess whether some of their current findings are specific to IFN β . Questions that could be addressed in this context include: Do similar preferential effects on HLA-B molecules and epitopes occur after treatment with IFN γ ? Is the impact of both cytokines on the immunopeptidome comparable? Do these effects correlate with changes in the transcriptome, or are they the result of post-translational alterations? Are there signaling differences in both molecules that could explain the observed distinctions? Is there any relevance to the pathophysiology of Type 1 Diabetes (T1D)? While the authors have briefly touched upon some of these considerations in the discussion section, providing data to support these points seems feasible and could significantly enhance the novelty of the study.

Reply: We agree with the Reviewer that the differences with our previous work were not clearly stated, which limits the appreciation of the novelty of our current report. The strengths and novelties of our current study are multiple.

First, there are major methodological improvements, using higher number of cells and optimized bioinformatics pipelines. Despite a higher stringency, this leads to an overall depth of peptide identification which is 6.3-fold higher compared to our previous work, which identified a total of 164 peptides as compared to the current 1,031 peptides. This is now shown in new Supplementary Figure 5a.

Second, the issue of neo-antigens (which we define as modified peptide sequences not templated in the genome, see below) was only marginally addressed. For instance, we have here used a novel MARS bioinformatics algorithm developed by Dr. Ternette (Liao, *Nat Commun* 2024) to search for possible *cis*-spliced peptides. Our previous approach was limited, as it simply relied on an extension of the reference proteome database used to interrogate the immunopeptidomics dataset with putative sequences based on published proteasomal preferences for *cis*-splicing events, thus increasing the risk of false positive discoveries. Our present strategy instead relies on *de-novo* sequencing and is less exposed to this risk. Moreover, post-translational modifications were not analyzed altogether in our previous work, while we here validated them under stringent experimental conditions, including comparison with spectra from native and modified peptides to exclude artifacts. This led to the identification of four post-translational modifications specifically detected in IFN- α -treated conditions, notably including glutathionylation.

Third, our present work addresses a different biological question, which is the increased immune visibility of β -cells induced by IFN- α and more generally by IFNs. This difference between the β -cell

immunopeptidome under basal and IFN-treated conditions was also marginally addressed in our previous work, due to more limited quantification tools and lower peptide yields.

Fourth, this deeper peptide identification and improved, automated bio-informatics pipeline relying on *de-novo* sequencing highlighted the preferential presentation of HLA-B ligands upon IFN- α treatment that previously went unnoticed. Following Reviewer's suggestion, we re-analyzed the previous dataset to look at HLA allele-specific features. Applying our new pipeline to our previous dataset, we identified a similar trend of preferential presentation of HLA-B ligands upon exposure to IFN- γ /IL-1 β and, to a lesser extent, to IFN- γ alone. Thus, this preference is not restricted to IFN- α , as expected based on the HLA-A vs. HLA-B upregulation studies presented in the manuscript. These results are now shown in new Supplementary Fig. 5b. Of note, this cytokine-dependent upregulation of HLA-B ligands previously went unnoticed with our previous analysis pipeline (see figure below), which adds value to the bioinformatics improvements introduced.

We have now summarized these multiple points of novelty in the revised Discussion (page 18), and we thank the Reviewer for drawing our attention on this omission.

Reviewer: - The term “neo-antigen” used in the title is not defined in the text. Sometimes the terms “neo-epitopes” and “neo-sequences” are also used, are they synonyms? I assume that the definition of neo-antigen is: “any peptide presented by IFN- α treated cells, and not presented by control cells”. In this case, the authors do not provide full details about the neo-antigens (which are yet one of the main topic of the article). They only mention that some (18) PTM or spliced peptides are neo-antigens. However, they also mention (line 132) that 35.6% of peptides are specifically presented by IFN- α treated cells. What is the composition of these neo-peptides? Do they derive from proteins that are supposedly immunogenic? Why are the PTM/spliced peptides so important to deserve a detailed/focused analysis while the vast majority of peptides is “ignored”? Are they supposed to be more immunogenic? Overall, it is not clear to me why these neo-antigens are special/interesting.

Reply: The term “neo-antigen” is a synonym for “neo-sequence”, and refers to non-canonical peptide sequences, i.e. not templated in the genome, resulting from post-translational modifications, peptide *cis*-splicing or mRNA splicing events. While we show that these neo-antigens largely arise upon cytokine exposure, this term does not indicate “any peptide presented by IFN- α treated cells”. We apologize for not making this clear – we have now amended this in the Introduction.

We respectfully argue that IFN- α -specific peptides are not ignored – they are actually quite extensively described and commented in the manuscript. Apart from the IFN- α -specific PTM pattern, a significant portion of the Results section covers IFN- α -specific differences in HLA Class I restriction and granule protein origin of the peptides identified. Additional investigations will be required to conclude on their immunogenicity, but this is out of the scope of the present study. We conducted a detailed study of PTM/spliced peptides because those are the sequences not templated in the genome, and may therefore be regarded as “non-self” (i.e. neo-antigens), thus preferentially escaping immune tolerance and favoring T-

cell recognition. Moreover, these modifications are often induced under inflammatory conditions in target tissues, thus favoring tissue-specific autoimmune recognition, including for those antigens without a tissue-specific expression pattern. This explains the growing interest in neo-antigens in the field of cancer and autoimmunity, and particularly in T1D (e.g. see review Purcell et al, *Diabetes* 2019). The prototypic example of the relevance of these neo-antigens is provided by the citrullinated antigens targeted by autoantibodies and T cells in rheumatoid arthritis. This however contrasts with the paucity of evidence in the literature for the natural processing and presentation of these neo-antigens, which we believe adds merit to our work. This point has now been clarified in the Introduction.

Reviewer: - Regarding neo-antigens deriving from PTM or splicing, the authors should consider that many genomic loci other than exons can generate MHC epitopes (PMID 26728094). Therefore, it is possible that the genomic origin of the neo-antigens that they identify is not the one they assume but could be a genomic sequence absent from their MS database or transcriptome mapping tools. To illustrate this, a recent study (PMID 37582761) has shown that many MHC epitopes assumed to derive from proteasomal cis-splicing (PMID 27846572, this study is not cited to my great surprise, while it is the founding study about peptide cis-splicing), can in fact derive from genomic-templated (non-spliced) loci. A possible verification analysis here would be to check the absence of a genomic locus capable of coding for their neo-antigens (or their leucin/isoleucine variants) with the website <https://bamquery.irc.ca/search> (the tool used in PMID 37582761)

Reply: We thank the Reviewer for raising this important point and for his/her valuable suggestion. Indeed, our previous version of the manuscript employed a prototype version of the MARS algorithm that did not consider these possibilities. We have now re-analyzed the putative cis-spliced sequences with the most recent and now published MARS implementation (Liao, *Nat Commun* 2024), which does consider possible alternative origins of putative cis-spliced peptides, notably non-canonical mRNA reading frames, including allegedly non-coding ones (Laumont, *Nat Commun* 2016). This reanalysis did not return any alternative assignment.

Moreover, a BamQuery analysis of these putative *cis*-spliced sequences did not retrieve any plausible alternative assignments to allegedly non-coding non-canonical mRNA reading frames (Laumont, *Nat Commun* 2016; Ruiz-Cuevas, *Genome Biol* 2023). This additional analysis is now mentioned in the Results section along with the relevant references.

Reviewer: - The rationale for the proteasome-related immunopeptidomic analyses is not clear to me, and the interest of the conclusion (lines 225-226) is not clear either. Is it not perfectly expected that the immunopeptidome is shaped by the activity of the constitutive- and immuno-proteasomes? This part seems unrelated to the rest of the manuscript. To reconnect these observations to the rest, the authors could check whether IFN α -induced peptide (neoantigens) derive more frequently from chymotryptic cleavage since IFN- α upregulates the immunoproteasome.

Reply: We thank the Reviewer for this excellent suggestion. We have performed the proposed analysis, which indeed show an increase in peptides carrying chymotrypsin cleavage motifs upon IFN- α exposure. These analyses are presented in new Fig. 2c-d and in a separated Results section. The flow of the paper has been revised accordingly and now makes this part more connected to the rest.

Reviewer: - My previous comment illustrates well a more general comment: many analyses seem to be only for descriptive purpose, and to be performed without a clear rationale/conclusion to be taken. This results in a manuscript that is hard to read and a general message that is diluted in observations that seem irrelevant to the main message. I would encourage the authors to make their manuscript more focused/straightforward.

Reply: We respectfully argue that the manuscript indeed starts with descriptive results, which are however instrumental to formulate a hypothesis, i.e. that IFN- α significantly skews the immunopeptidome display, and to subsequently thoroughly test it in the second part of the work. We apologize if this did not come through, we have now taken care to clarify this reasoning in the transition between the different Results sections.

Reviewer: - It is not clear how IFN α increases the expression of HLA-B without affecting dramatically the other alleles (HLA-A/C). The authors should offer a mechanism, supported by experimental data, to explain this.

Reply: The answers to this important question can be found in the literature. Both transcriptional and post-transcriptional mechanisms have been described. First, the HLA-B locus-specific IRE has a higher affinity for IRF-1 and -2 (Hakem, *J Immunol* 1991; Girdlestone, *PNAS* 1993). Second, ubiquitin-mediated degradation of HLA-A but not HLA-B mRNA may also be at play (Cano, *EMBO J* 2012). These mechanisms are now more thoroughly commented in the Discussion (page 18).

Reviewer: - Do the authors have an explanation to the immunogenicity of ABCC8 and CHGA-derived peptides? These are normal proteins (not mutated or dysfunctional) that are part of the normal biological processes of the organism. Their recognition by CD8 T cells is not trivial.

Reply: Indeed this is one of the key questions in the field of autoimmunity at large. The same process is at play for the main T1D autoantigen INS and for others in other autoimmune diseases. This is due to a breakdown of immune tolerance, which results from an interplay between genetic predisposition, environmental triggers, immune dysregulation and increased vulnerability of target tissues.

Reviewer: - This latter comment raises more general questions that could help the authors clarify/develop their manuscript:

A) Do the authors have an explanation for the origin of the immune reaction targeting HLA-B-presented peptides?

B) Is it expected that HLA-B overexpression by IFN-alpha automatically results in the generation of peptides recognized by CD8 T cells, or is this phenomenon restricted to individuals predisposed to T1D?

C) In case they assume that HLA-B overexpression leads to greater recognition of HLA-B-presented peptides by CD8 T cells in healthy individuals (response to previous question), do they believe this also applies to other HLA alleles?

D) If the response is yes in the previous question, how do they explain that normal peptides become immunogenic simply through the upregulation of the HLA molecule presenting them?

E) Finally, how is it possible that any anti-viral immune reaction (producing IFN α) in the pancreas does not induce autoimmune reactions against beta cells if IFN α induces HLA-B molecules, triggering the activation of CD8 T cells attacking beta cells?

Reply:

A) We do not have a definite explanation for this. With very few exceptions outside the T1D field (e.g. cross-recognition between the gliadin food antigen and the tissue transglutaminase self-antigen in celiac disease), a conclusive explanation is lacking for any autoimmune reaction against self-peptides, irrespective of their HLA restriction. Our hypothesis is that HLA-B-restricted peptides contribute marginally to the antigen display of beta cells under physiological conditions (so called immune ignorance). In the local inflammatory islet microenvironment of T1D, IFN- α may drive the preferential exposure of these HLA-B-restricted peptides and thus contribute to tolerance breakdown by promoting immune visibility. This is now mentioned in the Discussion (page 18) and a schematic of this hypothesis is presented in new Supplementary Fig. 11 (reproduced below).

B-C-D) This phenomenon is expected to be restricted to individuals predisposed to T1D because it relies on the IFN- α -driven local inflammation in pancreatic islets (also called insulinitis). Insulinitis is not present in individuals not predisposed to T1D.

E) Viral triggers, notably by Enteroviruses such as Coxsackievirus B, are actually plausible environmental candidates for T1D, and the IFN- α signature of the disease has been linked to these viral triggers (for recent reviews, see Nekoua, *Nat Rev Endocrinol* 2022; Carré, *Endocr Rev* 2023). It is thus well possible that viral infections reaching the pancreas (as is the case for Enteroviruses) may trigger the local IFN- α response that upregulates HLA-B and favors the autoimmune attack of CD8+ T cells against beta cells.

Reviewer: - Line 112, the authors indicate that their reference database has been complemented with translated mRNA isoforms enriched in human islets exposed or not to IFN α . It is not clear to me what are these isoforms and why they were absent from the reference database. Likewise, why are the open-reading frames of insulin not included in this reference database?

Reply: As explained in the Methods section, we used a reviewed human UniProt reference database, which only contains reviewed human canonical protein sequences (Swiss-Prot) and no mRNA isoform translation product. The mRNA isoforms that were appended to this database were translated from published RNAseq datasets derived from human islets exposed or not to IFN- α (Colli, *Nat Commun* 2020; GSE148058) and from previously reported INS DRiPs (Kracht, *Nat Med* 2017). As detailed in the Methods section, mRNA isoforms that were enriched in human islets and not expressed in mTECs were translated and appended to the canonical protein database.

Reviewer: - It is not clear how many mTECs were sequenced and used in the selection pipeline of neoantigens in line 608-621 (if this is indeed the selection filters for neoantigens, this is not clear either).

Reply: These mTECs came from 5 thymic samples, and these datasets were used in our previous work (Gonzalez-Duque et al, *Cell Metab* 2018). This work has been recently published more extensively by our collaborators on that paper (Carter et al, *Nat Commun* 2022; GSE201719) and is now quoted for reference. This is indeed one selection filter used for neo-antigens derived from mRNA splicing.

Reviewer: - I could not find the repository for mTEC RNA-seq data (fastq files), the authors should indicate where reviewers and readers can find these data to ensure the reproducibility of their pipeline.

Reply: These datasets are deposited under GSE201719, now referenced in the manuscript.

Reviewer: - The authors indicate on line 595 that “RNAs from 6 individual preparations of primary human islets exposed or not to IFN- α for 18h [...] were sequenced on an Illumina HiSeq 2000 at high depth” and on line 606 that “Datasets have been deposited under GEO:GSE148058.”. This suggests that the RNA sequencing and data deposition were performed for the need of the present study, by its main authors. However, the GSE148058 dataset was deposited by the lab of Maikel Luis Colli (co-author of this study) for the need of their study in 2020 (PMID 32444635). This is misleading then and should be rephrased to clearly acknowledge that the generation of the data was made in another study.

Reply: This has now been corrected in the Methods section.

REPLIES TO REVIEWER COMMENTS

Reviewer #1

Reviewer: A substantial amount of work has been put into this revision and I am grateful to the other for this effort. While the criticism about HC10, the main anti-HLA-B antibody used for validation in the study, has been partially answered, its exclusive use in tissue section is still short of alleviating all concerns. I had suggested the use of anti-HLA-B polyclonal antibodies for confirmation. I am saddened that this step has not been taken.

Reply: We thank the Reviewer for her/his appreciation of the revision work. We respectfully argue that 2 different HLA-B antibodies were used to support our conclusions:

1) HC10 for Western blot and tissue immunofluorescence, whose cross-reactivity has been previously extensively characterized (Perosa, *J Immunol* 2003), and we took care to exclude tissue donors harboring potentially cross-reactive HLA-A alleles to avoid any pitfall.

2) JOAN-1 for flow cytometry, which we extensively characterized against a large panel of mono-allelic HLA Class I K562 transductants (Supplementary Fig. 6), documenting marginal cross-reactivity with non-HLA-B alleles.

Moreover, the use of an anti-HLA-B polyclonal antibody, i.e. recognizing multiple epitopes, would increase rather than decrease the impact of such cross-reactivities.

Reviewer #5

Reviewer: Figure 1 has been clarified, as requested by Reviewer #2, and the addition of Discussion and Supplementary Fig.11 nicely highlight the potential pathomechanistic significance of the findings in the manuscript.

Reply: Thank you.

As for the request of adding additional evidence for the pathogenic role of islet peptides presented by HLA-B and a demonstration of killing potential of HLA-B40 restricted T cells, the authors have 1) performed additional analysis of islet-derived T-cell lines, demonstrating the recognition of multiple islet epitopes in the context of HLA-B40 and 2) indirectly demonstrated that IFN- α -induced upregulation of HLA-B expression leads to increased cytolytic activity using non-islet-specific TCRs and peptide-pulsed ECN90 cells. These expand on the original experiments, but naturally do not still conclusively demonstrate a pathogenic role of HLA-B epitopes in T1D pathogenesis. This would understandably require complex experiments that are beyond the scope of the current manuscript.

I would still suggest to add this caveat shortly in the Discussion, i.e., that the current study does not conclusively demonstrate that the increased presentation of HLA-B-restricted islet peptides leads to increased killing of beta cells and consequently further research would be needed to clarify whether the increased presentation of islet peptides by HLA-B has a potential pathogenic or protective effect on T1D pathogenesis.

Reply: We agree that a caveat is needed, and have added the following sentence in the in the “limitations” paragraph of the Discussion: “Third, further research is needed to conclusively demonstrate that the increased presentation of HLA-B-restricted endogenous peptides leads to increased β -cell killing.”